# Is Graph Mixup Beneficial? Investigating Interpolation And Empirical Performance of Graph Mixup Methods

## Abstract

Mixup is a widely used data augmentation technique that constructs new training examples by interpolating between existing ones. While simple and effective in domains like vision and language, applying mixup to graph data is non-trivial and independent empirical evidence for its effectiveness is lacking. To fill this gap, we conducted an independent evaluation following established evaluation protocols for graph classification and found that none of the state-of-the-art mixup methods yielded statistically significant improvements over the no-mixup baseline. To obtain further insights, we analyzed the graphs generated from existing mixup methods from an interpolation perspective using the graph edit distance. We found that (i) many mixup methods failed to interpolate well, (ii) high interpolation error led to performance degradation, and (iii) even optimal interpolation did not lead to performance improvements. Our findings highlight the need for a more rigorous exploration and evaluation of mixup for graphs.

## 1 Introduction

Data augmentation is an essential technique for improving generalization in machine learning and is particularly useful in domains where training data is scarce. Mixup (Zhang et al., 2018), a popular data augmentation technique, creates new training examples by *interpolating* between existing ones. Originally introduced in computer vision (Zhang et al., 2018), mixup has shown strong regularization and calibration effects (Thulasidasan et al., 2019) in other domains as well; e.g., in speech recognition (Meng et al., 2021) and natural language processing (Sun et al., 2020). Mixup is appealing due to its simplicity and intuitive design, its applicability without requiring domain-specific knowledge, and since it affects both the input and label space (in contrast to augmentation techniques such as DropNode (Do et al., 2021) or DropEdge (Rong et al., 2019)). The resulting soft labels encourage linear separation between classes in the model's representation space.

In this paper, we revisit mixup for graph classification tasks. Since graphs are complex and irregular, it is not immediately obvious how mixup should be performed. In the recent years, several alternative approaches for graph mixup have been proposed to address this question, drawing from domains such as optimal transport (Villani, 2008), graph theory (Lovász and Szegedy, 2006), and graph matching networks (Li et al., 2019). However, reproduction issues have been raised for some of the methods (Omeragic and Duranović, 2023), there are no independent evaluations of graph mixup methods, and the evaluations that have been performed often focused on empirical performance and did not analyze the produced mixup graphs directly.

**Main contributions (C) and results (R).** **(C1)** We address these gaps by first performing an independent evaluation of graph mixup in a unified experimental setup following established evaluation protocols (Errica et al., 2019). **(R1)** We found that graph mixup provided no significant improvement over the no-mixup baseline, which questions the practical benefits of graph mixup. **(C2)** We then performed a pooled analysis in weaker statistical setups. **(R2)** Even after pooling, graph mixup provided no significant improvement over the no-mixup baseline. **(C3)** To obtain further insights, we then systematically analyzed the mixup graphs produced by existing mixup methods from an interpolation perspective. **(R3)** We found that most graph mixup methods did not interpolate well, and that **(R4)** poor interpolation properties were detrimental for empirical performance.

Table 1: Overview of graph mixup methods

| Method | Interpolating? | Inputs | Alignment | Output | Learned? |
|---|---|---|---|---|---|
| Embedding Mixup | ✓ | Embeddings | N.A. | Embedding | ✗ |
| FGW-Mixup | ✓ | Graphs | FGW coupling | Graph | ✗ |
| If-Mixup | ✓ | Adjacency matrices | Arbitrary | Edge-weighted graph | ✗ |
| G-Mixup | ✓ | Graphons | Degree | Graphon | ✓ |
| GeoMix | ✓ | Graphs | GW coupling | Edge-weighted graph | ✗ |
| MomentMixup[1] | ✓ | Motif densities | Motifs | Graphon | ✓ |
| S-Mixup | ✓ | Adjacency matrices | Learned | Edge-weighted graph | ✓ |
| SIGL[1] | ✓ | Graphons | Learned | Graphon | ✓ |
| SubMix | ✓ | Graphs | Random | Graph | ✗ |
| GED-Mixup (Sec. 4.3) | ✓ | Graphs | Optimal | Graph | ✗ |

## 2 GRAPH MIXUP

Mixup was originally introduced by Zhang et al. (2018) as a data augmentation technique for supervised learning tasks, particularly in computer vision and speech recognition. In this paper, we study mixup for supervised graph classification, where the inputs are graphs (potentially including node/edge features) and the goal is to learn a classifier for unseen graphs from a set of labeled examples. Application areas include the biomedical data (Qabel et al., 2022; Wang et al., 2025; Buterez et al., 2024), bioinformatics (Jang et al., 2024; van der Weg et al., 2025; Jha et al., 2022)), cybersecurity (Bilot et al., 2024), fraud detection (Motie and Raahemi, 2024), and many more (Cao et al., 2024; Park et al., 2022; Jin et al., 2025). Training data is often scarce in these applications so that mixup is a promising approach to combat overfitting.

**Linear mixup.** Zhang et al. (2018) considered inputs and labels represented as real-valued tensors (e.g., an image and its one-hot encoded class label) and performed mixup by linear interpolation. More precisely, given two input examples $(x_1, y_1)$ and $(x_2, y_2)$, their *linear mixup* approach constructs a synthetic example $(x_M, y_M)$ by taking a convex combination of both inputs and labels

$$x_M = x_1 + \lambda(x_2 - x_1) = (1 - \lambda)x_1 + \lambda x_2 \quad \text{and} \quad y_M = (1 - \lambda)y_1 + \lambda y_2, \tag{1}$$

where $\lambda \in [0, 1]$ refers to a *mixup ratio*. Intuitively, $\lambda$ describes how far the result moves away from the first towards the second input. Linear mixup improved generalization and robustness in their experimental study, and mixup methods have been widely adopted and extended since then (Shamsian et al., 2024; Yun et al., 2019; Touvron et al., 2021; Liu et al., 2021; Verma et al., 2019; Ramé et al., 2021; Bao et al., 2023; Zou et al., 2023).

**Graph mixup.** Graph mixup methods formulate graph mixup—in the spirit of linear mixup—as *interpolation* between two example graphs and their labels. Conceptually, these methods interpolate the class labels of their inputs using Eq. (1), but differ in how they interpolate between the input graphs themselves. Intuitively, most methods use *alignments* to determine "common parts" between the two input graphs (Fig. 1, top right, common parts color-coded). Mixup is then performed only on the "different parts" by including nodes and edges from both graphs proportional to the desired mixup ratio (see "good mixup" in Fig. 1, $\lambda = 40\%$). This is similar to linear mixup, in which common elements of the input tensors are left unchanged and different elements are subject to mixup. See App. A.1 for a discussion of why graph mixup can be beneficial, and App. A.2 for a discussion of graph mixup compared to other augmentation methods.

**Graph mixup methods.** Several graph-specific mixup methods have been introduced in the literature, including If-Mixup (Guo and Mao, 2023), S-Mixup (Ling et al., 2022), SubMix (Yoo et al., 2022), G-Mixup (Han et al., 2022), MomentMixup[1] (Ramezanpour et al., 2025) and SIGL[1] (Azizpour et al., 2025), FGW-Mixup (Ma et al., 2023), GeoMix (Zeng et al., 2024), and Embedding Mixup (Wang et al., 2021). Key differences between these methods include: (i) the inputs to mixup, (ii) how these inputs are aligned, (iii) the output of mixup, and (iv) whether or not the mixup method itself is learned. A brief overview along these dimensions is given in Tab. 1; see also App. B.

---

[1] MomentMixup and SIGL were published concurrently to this work. Both methods follow G-Mixup in that they estimate graphons and subsequently sample mixup graphs. We do not consider these methods further in this study and leave an independent evaluation to future work.

**Inputs to mixup.** Mixup can be performed on (i) two graphs, (ii) two adjacency matrices, (iii) two graphons (each representing the set of graphs associated with a class label), or (iv) two graph embeddings produced by the downstream network (along the lines of Manifold Mixup (Verma et al., 2019)). For (ii)–(iv), interpolation is typically done using linear mixup after suitable preprocessing (e.g., reordering adjacency matrices according to an alignment and adding singleton nodes to match their sizes), whereas (i) is handled differently. In particular, SubMix (Yoo et al., 2022) adopts a strategy inspired by CutMix (Yun et al., 2019): given two input graphs, it replaces a subgraph of one input with a subgraph of the other input. FGW-Mixup (Ma et al., 2023) and GeoMix (Zeng et al., 2024) rely on the (Fused-)Gromov-Wasserstein distance (Vayer et al., 2020) from the theory of optimal transport (Villani, 2008). While the former method aims to compute barycenters, the latter relies on geodesics (Peyré et al., 2016). In both cases, mixup takes place in the Gromov-Wasserstein space and is computationally expensive so that approximation algorithms are used.

**Alignments.** Most graph mixup methods (implicitly or explicitly) make use of an alignment between their inputs. Obtaining a good alignment can be challenging and computationally expensive (Chang et al., 2023). Different approaches have been explored: (i) arbitrary (i.e., determined by how the graphs happen to be provided), (ii) learned, (iii) random, (iv) degree-based ordering, (v) a coupling in the sense of optimal transport (Villani, 2008). We also explore (vi) an optimal alignment in this paper (for analysis; see Sec. 4.3). Note that Embedding Mixup (Wang et al., 2021) applies mixup on graph embeddings; here the notion of alignment is not directly applicable.

**Outputs.** Mixup methods can produce as output: (i) a mixup graph, (ii) an edge-weighted mixup graph, (iii) a graphon, and (iv) embeddings. Here (i) stays in the input space, (ii) produces graphs in which edges are labeled with "existence probabilities", (iii) can be used to sample mixup graphs, and (iv) stays in the embedding space of the downstream network.

**Learned mixup.** S-Mixup (Ling et al., 2022) and G-Mixup (Han et al., 2022) use learned mixup, i.e., they need to be trained on the training data used for the downstream task beforehand.

## 3 EMPIRICAL ANALYSIS

We performed an independent experimental study to evaluate the empirical performance of state-of-the-art graph mixup methods for graph classification in a common setup. Our key goals were to assess to what extent graph mixup is beneficial in that it increases prediction performance. Our code and results will be provided via GitHub for the full submission.

**Experimental setup.** We independently evaluated on six representative datasets from TUDataset (Morris et al., 2020) commonly used for graph classification tasks (see App. C.1 for dataset statistics and our rationale for choosing datasets), using GCN (Kipf and Welling, 2017) and GIN (Xu et al., 2018) as backbone models. Our methodology followed the careful choices of Errica et al. (2019), i.e., nested cross-validation for model selection and assessment, repeated runs for robustness, and significance testing via Welch's $t$-test (Welch, 1947). We report mean test accuracy with standard errors across 5-fold nested CV, with three repetitions per split. We considered seven graph mixup variants of Tab. 1 along with GED-Mixup (a baseline introduced in Sec. 4.3). Mixup graphs were randomly generated in each training epoch. Hyperparameters—including model, training, and mixup settings—were tuned individually for each dataset/method via random search plus Bayesian optimization with TPE (Bergstra et al., 2011) using Optuna (Akiba et al., 2019). The experimental setup is described in detail in App. C.2.

**Result ①: Graph mixup provided no significant improvement over the no-mixup baseline.** Tab. 2 shows the results for all methods evaluated on GCN (Kipf and Welling, 2017) and GIN (Xu et al., 2018), respectively. Although some mixup methods appear to improve over the no-mixup baseline, none of these results are statistically significant. Some methods significantly fell behind the no-mixup baseline. These results question whether mixup is beneficial for graph classification tasks.

**Pooled analysis.** To provide more insight, we performed a pooled analysis in multiple statistical setups; our results are given in Tab. 3. We pooled over models, datasets, folds, and/or runs and report average accuracy as well as standard error. The estimates of the standard error depends on the underlying assumptions. In particular, we make three assumptions of increasing strength (and hence decreasing standard error estimates):

Table 2: Test accuracy (%) and standard error (pp) for multiple datasets. Missing entries indicate cases where FGW-Mixup could not generate mixup graphs. Statistically significant differences over the no-mixup baseline are marked bold (associated $p$-values in App. C.6). Note that we did not correct for multiple testing.

| Method | MUTAG | ENZYMES | IMDB-B | PROTEINS | IMDB-M | NCI1 |
|---|---|---|---|---|---|---|
| | | | GCN | | | |
| Baseline | $78.42 \pm 1.71$ | $72.56 \pm 1.21$ | $68.80 \pm 1.00$ | $71.83 \pm 2.94$ | $46.31 \pm 0.60$ | $80.35 \pm 0.57$ |
| Emb-M. | $79.66 \pm 1.45$ | $70.89 \pm 1.21$ | $66.60 \pm 1.85$ | $70.09 \pm 2.13$ | $\mathbf{43.18 \pm 1.03}$ | $81.63 \pm 0.29$ |
| FGW-M. | $76.82 \pm 2.34$ | – | – | – | – | – |
| G-Mixup | $82.48 \pm 1.33$ | $68.28 \pm 2.02$ | $69.40 \pm 1.13$ | $74.13 \pm 1.05$ | $45.73 \pm 0.92$ | $80.83 \pm 0.36$ |
| GeoMix | $75.63 \pm 3.43$ | $74.00 \pm 1.19$ | $\mathbf{62.67 \pm 2.23}$ | $71.65 \pm 2.61$ | $\mathbf{44.40 \pm 0.47}$ | $80.98 \pm 0.31$ |
| If-Mixup | $81.43 \pm 1.56$ | $72.06 \pm 0.98$ | $68.53 \pm 1.11$ | $74.88 \pm 1.06$ | $45.47 \pm 0.73$ | $81.43 \pm 0.36$ |
| S-Mixup | $80.21 \pm 1.67$ | $67.29 \pm 5.38$ | $70.06 \pm 1.61$ | $72.26 \pm 1.97$ | $\mathbf{40.58 \pm 1.72}$ | $79.39 \pm 2.44$ |
| SubMix | $80.03 \pm 1.88$ | $73.39 \pm 1.42$ | $68.20 \pm 1.27$ | $72.87 \pm 2.17$ | $45.20 \pm 1.03$ | $81.48 \pm 0.38$ |
| GED-M.[1] | $81.64 \pm 1.81$ | $72.11 \pm 1.39$ | $68.97 \pm 1.23$ | $73.44 \pm 1.39$ | $46.16 \pm 0.76$ | $81.58 \pm 0.25$ |
| | | | GIN | | | |
| Baseline | $84.41 \pm 1.39$ | $70.33 \pm 0.98$ | $70.77 \pm 0.53$ | $69.91 \pm 3.45$ | $48.09 \pm 0.58$ | $81.65 \pm 0.42$ |
| Emb-M. | $81.89 \pm 1.34$ | $70.78 \pm 1.02$ | $67.77 \pm 1.89$ | $71.26 \pm 2.59$ | $48.22 \pm 0.64$ | $81.96 \pm 0.42$ |
| FGW-M. | $82.81 \pm 1.42$ | – | – | – | – | – |
| G-Mixup | $80.49 \pm 1.75$ | $69.17 \pm 1.11$ | $65.90 \pm 2.38$ | $68.62 \pm 3.40$ | $47.02 \pm 1.35$ | $80.84 \pm 0.57$ |
| GeoMix | $81.78 \pm 2.23$ | $69.00 \pm 1.31$ | $70.53 \pm 0.60$ | $69.46 \pm 2.96$ | $47.49 \pm 0.77$ | $81.74 \pm 0.40$ |
| If-Mixup | $84.09 \pm 1.39$ | $70.06 \pm 1.38$ | $69.30 \pm 0.72$ | $70.12 \pm 3.51$ | $47.91 \pm 0.57$ | $81.74 \pm 0.34$ |
| S-Mixup | $\mathbf{80.83 \pm 0.90}$ | $68.96 \pm 1.35$ | $69.29 \pm 2.16$ | $62.37 \pm 2.24$ | $46.84 \pm 0.70$ | $79.25 \pm 1.30$ |
| SubMix | $84.75 \pm 1.64$ | $70.72 \pm 1.43$ | $70.40 \pm 0.45$ | $71.08 \pm 2.70$ | $47.96 \pm 0.60$ | $82.21 \pm 0.44$ |
| GED-M.[1] | $82.84 \pm 1.35$ | $71.17 \pm 0.95$ | $70.40 \pm 0.76$ | $68.30 \pm 3.35$ | $47.89 \pm 0.88$ | $82.04 \pm 0.48$ |

[1] Introduced in Sec. 4.3.

**A1** (standard): Treat datasets and model classes as sampled from a dataset and model class distribution. This allows us to make statements about empirical performance on new datasets and model classes, which is what we are ultimately interested in. This is the weakest assumption considered in this study.

**A2** (fixed distribution): Treat datasets and model class as sampled from fixed data and model distributions. This allows to make statements about empirical performance of GCN/GIN when applied to the data distribution underlying our evaluation datasets.

**A3** (fixed data): Treat the evaluation data as the entire population (i.e., treat the empirical distribution as the data distribution). This allows us to make statements about the particular data that is used (but not about their underlying data distribution). This is the strongest assumption considered in this study.

A more thorough discussion and details are provided in App. D.

**Result ②: Even after pooling, graph mixup provided no significant improvement over the no-mixup baseline.** Our pooled results are shown in Tab. 3. First, under (A1) none of the results were significant even after pooling. For (A2) and (A3) we obtained statistical significance in that some methods performed worse than the no-mixup baseline. So even in the most generous point of view (A3), we did not obtain statistically significant results in favor of mixup. Reasons for this negative result include the suitability of graph mixup in general, potential flaws in the mixup methods, or insufficient power (e.g., due to small effect sizes).

## 4 MIXUP AS INTERPOLATION

To investigate the failure of all considered graph mixup methods to produce significant performance benefits over the no-mixup baseline in our experiments, we now take a closer look at the generated mixup graphs. Recall from Sec. 2 that the goal of graph mixup is to interpolate between input graphs, according to a pre-specified mixup ratio $\lambda$. In this section, we formalize this interpolation goal and

Table 3: Pooled average test accuracy (%), pooled standard errors (pp) and $p$-values for GCN/GIN and the evaluation datasets under assumptions (A1)–(A3). FGW-Mixup is excluded due to missing results for some datasets. Statistical significant results over the no-mixup baseline are marked in bold.

| Method | Accuracy | A1 (standard) | | A2 (fixed dist.) | | A3 (fixed data) | |
|---|---|---|---|---|---|---|---|
| | | SE | p | SE | p | SE | p |
| Baseline | 70.29 | ±1.04 | – | ±0.46 | – | ±0.30 | – |
| Emb-M. | 69.49 | ±1.07 | 0.60 | ±0.43 | 0.21 | ±0.30 | 0.06 |
| G-Mixup | 69.41 | ±1.07 | 0.56 | ±0.48 | 0.19 | ±0.29 | **0.03** |
| GeoMix | 69.11 | ±1.10 | 0.44 | ±0.54 | 0.10 | ±0.27 | **<0.01** |
| If-Mixup | 70.58 | ±1.06 | 0.84 | ±0.40 | 0.63 | ±0.25 | 0.44 |
| S-Mixup | 68.11 | ±1.30 | 0.19 | ±0.66 | **0.01** | ±0.49 | **<0.01** |
| SubMix | 70.69 | ±1.07 | 0.79 | ±0.42 | 0.52 | ±0.24 | 0.29 |
| GED-Mixup (Sec. 4.3) | 70.54 | ±1.04 | 0.86 | ±0.42 | 0.68 | ±0.19 | 0.46 |

propose *interpolation error* metrics to quantify to what extent a mixup graph actually interpolates between inputs. To the best of our knowledge, such an analysis has not been done before.

In Sec. 5, we will use these results to study relationship between interpolation properties and empirical performance of graph mixup methods. We also consider an approach called GED-Mixup that interpolates optimally according to our metrics. While this method may not be practically viable in some applications due to its high computational costs, it provides a baseline result for the performance that optimal interpolation can achieve.

### 4.1 INTERPOLATION CRITERIA AND INTERPOLATION ERROR

We first make the intuition of "interpolation" more precise. Consider two inputs $x_1$ and $x_2$, a mixup ratio $\lambda \in [0, 1]$, and a mixup result $x_M$. Given a distance metric $d(\cdot, \cdot)$ between inputs, we say that $x_M$ $(d, \lambda)$-*interpolates* between $x_1$ and $x_2$ if the following *interpolation criteria* (IC)

$$\text{IC}_1: d(x_M, x_1) = \lambda \cdot d(x_1, x_2) \quad \text{and} \quad \text{IC}_2: d(x_M, x_2) = (1 - \lambda) \cdot d(x_1, x_2) \qquad (2)$$

are satisfied. We refer to the right-hand sides of $\text{IC}_1$ and $\text{IC}_2$ as *interpolation targets*. If both targets are met, then (i) $d(x_1, x_2) = d(x_1, x_M) + d(x_M, x_2)$ so that $x_M$ lies on a shortest path between $x_1$ and $x_2$ (w.r.t. $d$ and by the triangle equality) and (ii) the position of $x_M$ on this shortest path precisely matches the desired relative contribution of $x_1$ (i.e., $1 - \lambda$) and $x_2$ (i.e., $\lambda$).

To gain some intuition, observe that linear mixup of Eq. (1) produces the unique point that satisfies IC w.r.t. the Euclidean distance $d_2(x_1, x_2) = \|x_1 - x_2\|_2$ (and others, see Prop. 1 below), for both inputs and labels. This is visualized in Fig. 1 (left). Here $x_M$ is the result of linear mixup and $(d_2, 2/5)$-interpolates between $x_1$ and $x_2$. In contrast, $x_P$ satisfies only $\text{IC}_1$ but not $\text{IC}_2$. As this example highlights, both criteria are needed.

As we will see later, graph mixup methods often do not satisfy IC exactly but only approximately. To quantify the approximation error, we introduce the *absolute interpolation error (AIE)* given by

$$\text{AIE}_d(x_M; x_1, x_2, \lambda) \stackrel{\text{def}}{=} \big| d(x_M, x_1) - \lambda \cdot d(x_1, x_2) \big| + \big| d(x_M, x_2) - (1 - \lambda) \cdot d(x_1, x_2) \big|.$$

For brevity, we often write $\text{AIE}(x_M) = \text{AIE}_d(x_M; x_1, x_2, \lambda)$ and consider the remaining quantities as arbitrary but fixed. Observe that if $x_M$ $(d, \lambda)$-interpolates between $x_1$ and $x_2$, then $\text{AIE}(x_M) = 0$ (e.g., $x_M$ in Fig. 1). If it does not, then $\text{AIE}(x_M) > 0$ (e.g., $x_P$ in Fig. 1 w.r.t. $d_2$). Intuitively, the AIE measures the distance of the mixup result to actual $(d, \lambda)$-interpolation targets (cf. Fig. 1). As this distance can be arbitrarily large, the AIE is not bounded from above.

To be able to compare interpolation errors across input pairs $(x_1, x_2)$ with different distances $d(x_1, x_2)$ in a meaningful way, we normalize AIE w.r.t. $d(x_1, x_2)$ to obtain the mixup *interpolation error (IE)*:

$$\text{IE}_d(x_M; x_1, x_2, \lambda) \stackrel{\text{def}}{=} \frac{\text{AIE}_d(x_M; x_1, x_2, \lambda)}{d(x_1, x_2)} = \left| \frac{d(x_M, x_1)}{d(x_1, x_2)} - \lambda \right| + \left| \frac{d(x_M, x_2)}{d(x_1, x_2)} - (1 - \lambda) \right|. \quad (3)$$

The following proposition states that linear mixup interpolates optimally.

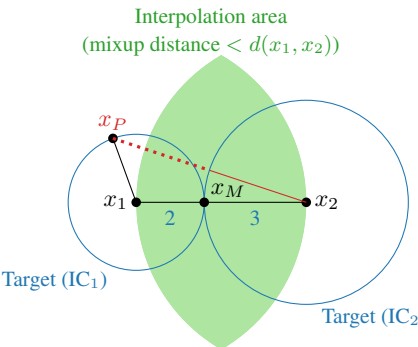 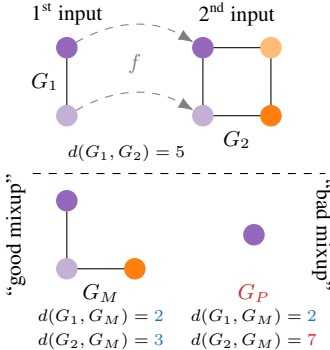

Figure 1: *Left:* $x_M$ $(d_2, 2/5)$-interpolates between inputs $x_1$ and $x_2$, whereas $x_P$ does not. The dotted line indicates $x_P$'s absolute interpolation error (AIE). *Right:* Similarly, $G_M$ $(d_{\text{GED}}, 2/5)$-interpolates between $G_1$ and $G_2$, whereas $G_P$ does not. We have $\text{AIE}(G_P) = |2 - 2| + |7 - 3| = 4$ and $\text{IE}(G_P) = 4/5$. The colors symbolize different node attributes. The mapping $f : V_1 \rightarrow V_2$ constitutes a *vertex mapping* or *alignment* between $G_1$ and $G_2$.

**Proposition 1.** *For any distance metric $d(a, b) = \|a - b\|$ induced by a norm $\|\cdot\|$ on the input/label vector space (over $\mathbb{R}$), linear mixup of Eq. (1) satisfies IC for inputs/labels and we have*

$$\text{AIE}_d(x_M) = \text{IE}_d(x_M) = 0 \quad / \quad \text{AIE}_d(y_M) = \text{IE}_d(y_M) = 0.$$

Such distances include, for example, the Manhattan distance (induced by 1-norm) and the Euclidean distance (induced by 2-norm). See App. F.1 for a proof.

## 4.2 Alignments, Edit Sets, and the Graph Edit Distance

As discussed in Sec. 2, graph mixup methods rely on alignments to produce mixup graphs. We now formalize the notion of an alignment, describe how alignments relate to edits sets and graph mixup, and finally define an optimal alignment based on the graph edit distance.

**Alignments.** Let $G_1 = (V_1, E_1)$ and $G_2 = (V_2, E_2)$ be two graphs where $|V_1| \leq |V_2|$ w.l.o.g. (otherwise swap $G_1$ and $G_2$). Intuitively, an alignment assigns to each node $v \in V_1$ a unique corresponding node $u \in V_2$. We formalize an alignment as an *injective vertex mapping* $f : V_1 \rightarrow V_2$; see Fig. 1 (top right) for an example. Recall that mixup methods aim to retain the "common parts" (sub-structures such as nodes, edges, or subgraphs) that exist in both input graphs and to mixup the remaining "different parts." An alignment formalizes what is meant by "common parts": for every node $v \in V_1$ its corresponding node $f(v) \in V_2$, and for every edge $(v_1, v_2) \in E_1$ its corresponding edge $(f(v_1), f(v_2)) \in E_2$ (if present). In the example of Fig. 1, the common part is given by the two purple nodes and the edge connecting these nodes.

**Edit sets.** Given an alignment, we can perform mixup by editing $G_1$ to bring it to closer towards $G_2$. To do so, we make use of *edit operations*. An edit operation is a node or edge insertion, deletion, or substitution (i.e., changing features). Let $\mathcal{F}(G_1, G_2)$ denote the set of all *edit sets*—i.e., sets of edit operations—that transform $G_1$ into $G_2$,[2] i.e.,

$$\mathcal{F}(G_1, G_2) = \big\{ F = \{e_1, e_2, \ldots, e_{|F|}\} : \text{apply}(G_1, F) \cong G_2 \big\},$$

where $e_i$ denotes an edit operation and $\text{apply}(G_1, F)$ denotes the result of applying all edit operations in $F$ to $G_1$. For any pair of graphs, there is an infinite number of edit sets that transform one into the other. An alignment $f$ induces a particular edit set $F_f \in \mathcal{F}(G_1, G_2)$, which only contains the edit operations for the "different parts." The edit set induced by $f$ in Fig. 1 contains five operations: two node insertion operations (for the orange nodes) and three edge insertion operations (for the edges

---

[2]Strictly speaking, $G_1$ is transformed into another graph $G_2'$ that is isomorphic to $G_2$, denoted by $G_2' \cong G_2$.

---

**Algorithm 1** GED-Mixup

---

**Require:** Graphs $G_1, G_2$; mixup ratio $\lambda \in [0, 1]$
**Ensure:** Mixup graph $G_M$
    $F^* \leftarrow$ a minimal edit set from $G_1$ to $G_2$ (i.e., $|F^*| = d_{\text{GED}}(G_1, G_2)$)
    $P \leftarrow$ a valid ordering of the edit operations in $F^*$ (e.g., chosen at random)
    $P_\lambda \leftarrow$ the first $\text{round}(\lambda|P|)$ edit operations in $P$
    **return** $G_M = \text{apply}(G_1, P_\lambda)$

---

incident to these nodes). Given $f$, edit set $F_f$ is cheap to obtain, i.e., in asymptotically linear time with respect to the number of nodes and edges (see App. B and Chang et al. (2023)).[3]

Given an edit set, we can perform mixup by applying a $\lambda$-fraction of the operations to $G_1$ (see Sec. 4.3 for details). In Fig. 1, mixup graph $G_M$ has been generated in this fashion with $\lambda = 2/5$ (and hence using 2 out of the 5 edit operations).

**Optimal alignments and graph edit distance.** An alignment is *optimal* if its induced edit set is as small as possible. Intuitively, this means that the alignment identifies a large common part. Alignment $f$ of Fig. 1 is such an optimal alignment. The size of the edit set induced by optimal alignment is given by the *graph edit distance (GED, Sanfeliu and Fu (1983), Chang et al. (2023))*:

$$d_{\text{GED}}(G_1, G_2) = \min_{F \in \mathcal{F}(G_1, G_2)} |F|.$$

In what follows, we write $d(G_1, G_2) = d_{\text{GED}}(G_1, G_2)$ for brevity. An example is given in Fig. 1.

Generally, computing the optimal alignment and/or GED is an NP-hard problem (Zeng et al., 2009). In fact, graph mixup methods typically do *not* use optimal alignments (cf. Tab. 1), and hence may produce problematic mixup graphs. For example, graph $G_P$ in Fig. 1 does not interpolate well between $G_1$ and $G_2$; graph $G_P$ has larger distance to $G_2$ than $G_1$ has to $G_2$ so that it does not interpolate at all. We further explore such questions in Sec. 5.

**Interpolation error.** The graph edit distance is the natural choice to quantify the interpolation error of mixup graphs using Eq. (3). In fact, most (all but Embedding Mixup) graph mixup methods implicitly make use of an alignment and its corresponding induced edit set. With this choice, the "good mixup" graph $G_M$ ("bad mixup" graph $G_P$) of Fig. 1 has interpolation error of 0 (4/5).

**Computational cost.** Even though GED computation is NP-hard, its computation can be feasible in practice. This is due to the availability of high-performance algorithms (e.g., Chang et al. (2020; 2023)) and since in our setting of graph classification, the graphs are comparably small (e.g., see dataset statistics in App. C.1). In our experimental study, we did not run into computational bottlenecks (e.g., see runtime results in App. C.3). We modified the code of AStar-BMao (Chang et al., 2023), a state-of-the-art algorithm for exact GED computation, such that it additionally yielded an optimal alignment $f^*$ and its induced edit set $F^*$—i.e., $|F^*| = d(G_1, G_2)$—as a by-product without any significant additional compute cost.

### 4.3 A BASELINE METHOD: GED-MIXUP

Most graph mixup methods rely on alignments when interpolating graphs. This raises the natural question of whether or not using an optimal alignment (instead of an approximate one) would benefit graph mixup. To investigate this question, we construct a simple baseline method—coined GED-Mixup—which uses the optimal alignment and serves as an analysis tool to study its effect on empirical performance. Without including GED-Mixup in our analyses, the effect of optimal interpolation on empirical performance would remain unclear. The method can be seen as a simplification of EPIC (Heo et al., 2024).[4]

---

[3]For example, compare the label of each $v \in V_1$ with the label of $f(v) \in V_2$ and add a substitution operation when they differ. As another example, for each edge $(v_1, v_2) \in E_1$, check whether $(f(v_1), f(v_2)) \in E_2$ present and add an edge insertion operation otherwise.

[4]EPIC uses learned cost models and GED approximations whereas our approach simply uses unit edit costs and exact GED. We did not consider EPIC in our experimental study as there is no source code available.

The method is described briefly in Alg. 1 and in more detail in App. E. It first computes a minimal edit set, orders the edit operation in the set, and the applies a fraction of $\lambda$ of the edit operations to $G_1$. We only consider *valid orderings*, in which (i) an edge can only be inserted when its source and target node are present, (ii) a node can only be removed when it does not have an incident edge, (iii) label substitutions are only possible for nodes/edges present in the graph, and (iv) when both $G_1$ and $G_2$ are connected, so is $G_M$. This approach avoids undesirable mixup results.

The following proposition shows that GED-Mixup is optimal in that its interpolation error is as small as possible (in particular, 0 whenever interpolation targets are integer). A proof is given in App. F.2.

**Proposition 2.** *GED-Mixup (Alg. 1) interpolates optimally w.r.t.* $d_{GED}$*, and it holds*

$$\mathrm{AIE}(G_M) = 2 \cdot |\mathrm{round}(\lambda \cdot d_{GED}(G_1, G_2)) - \lambda \cdot d_{GED}(G_1, G_2)| \leq 1.$$

We use GED-Mixup purely an *an analysis tool*, but do not consider it a competitive graph mixup method. The inclusion of GED-Mixup into our analyses allowed us to analyze the effect of optimal interpolation on empirical performance in graph classification tasks, which would remain unclear without this method. While GED computation can be expensive, using GED-Mixup was feasible in our study (see App. C.3).

## 5 INTERPOLATION ANALYSIS

Using the analysis tools AIE, IE, and GED-Mixup, we empirically investigate in this section how well existing mixup methods interpolate and how this related to their performance.

**Experimental setup.** We follow the experimental setup described in Sec. 3, but only considered methods that produce mixup graphs, i.e., SubMix (Yoo et al., 2022), If-Mixup (Guo and Mao, 2023), S-Mixup (Ling et al., 2022), GeoMix (Zeng et al., 2024), FGW-Mixup (Ma et al., 2023), and GED-Mixup.[5] During training, we collected all generated mixup graphs as well as their inputs and corresponding value of $\lambda$.[6] We used this approach because it allows us to analyze the mixup graphs actually used during training, and because some mixup methods are learned based on training data. Given a set $T = \left\{ (G_1, G_2, G_M, \lambda)_i \right\}_{i=1}^{n}$ of mixup graphs, we report the *mean interpolation error (mIE)* given by

$$\mathrm{mIE}(T) = \frac{1}{n} \sum_{(G_1, G_2, G_M, \lambda) \in T} \mathrm{IE}_{d_{\mathrm{GED}}}(G_M; G_1, G_2, \lambda).$$

An mIE value of 0 indicates that all mixup graphs perfectly interpolate between their inputs; larger values indicate larger errors. Values greater than 2 generally indicate particularly poor interpolation properties. To see this, observe that simply setting $G_M = G_1$ for an input pair leads to $\mathrm{IE} \leq 2$ (independently of $\lambda$); hence this clearly flawed approach already leads to an $\mathrm{mIE} \leq 2$ when used throughout training.

**Result ③: Most graph mixup methods did not interpolate well.** We sampled 500 mixup graphs for each combination of method, dataset, and choice of $\lambda \in [0.5 \pm \varepsilon], [0.8 \pm \varepsilon], [0.9 \pm \varepsilon]$ for $\varepsilon = 0.005$. Our results are summarized in Tab. 4; more detailed results are given in App. G.1. As can be seen, only GED-Mixup and SubMix generally produced graphs that interpolated well. This is expected for GED-Mixup, since it interpolates optimally by design. SubMix uses random alignments, but uses a more coherent CutMix approach (i.e., swap entire subgraphs) and thus is less impacted by sub-optimal alignments. S-Mixup, Geo-Mix, and FGW-Mixup did not produce mixup graphs that interpolated between their inputs, and If-Mixup fell in between. This suggests that the (approximate) alignments being used by the latter methods are far from optimal; we provide more detail below.

**Result ④: Poor interpolation properties were detrimental for empirical performance.** We now study to what extent interpolation properties correlate with empirical performance. Fig. 2 summarizes

---

[5]Methods that produce weighted mixup graphs, in which each edge is annotated with an "existence probability", are marked with $*$. To treat these methods appropriately when computing mIE, we account for edge weights by sampling edges according to their probability. Corresponding mIE scores can hence be interpreted as an "expected mIE". Accounting for edge weights in this fashion always decreased the corresponding mIE scores substantially.

[6]As $\lambda$-values are sampled from a Beta distribution during training or were out of our control (in SubMix), we allow for a small tolerance of $\varepsilon = 0.005$.

Table 4: Comparison of mean interpolation error (mIE) obtained by various methods and datasets (lower is better and $\geq 2$ is particularly poor). We were unable to generate graphs with FGW-Mixup on some datasets (denoted with –). Emb-Mixup and G-Mixup do not appear here as we only considered methods that perform pairwise mixup of two examples (required by mIE). More details can be found in App. G.1.

| Method | MUTAG | ENZYMES | IMDB-B. | PROTEINS | IMDB-M. | NCI1 | Avg. |
|---|---|---|---|---|---|---|---|
| GED-M. | 0.05 | 0.01 | 0.01 | 0.01 | 0.03 | 0.03 | 0.02 |
| SubMix | 0.45 | 0.28 | 0.49 | 0.30 | 0.53 | 0.37 | 0.40 |
| If-Mixup | 1.57 | 0.69 | 0.86 | 1.00 | 0.31 | 1.03 | 0.91 |
| S-Mixup | 3.56 | 0.95 | 1.32 | 0.72 | 0.96 | 2.10 | 1.60 |
| GeoMix | 4.31 | 1.32 | 0.66 | 1.60 | 0.69 | 2.60 | 1.86 |
| FGW-Mixup | 2.76 | – | – | – | – | – | 2.76 |

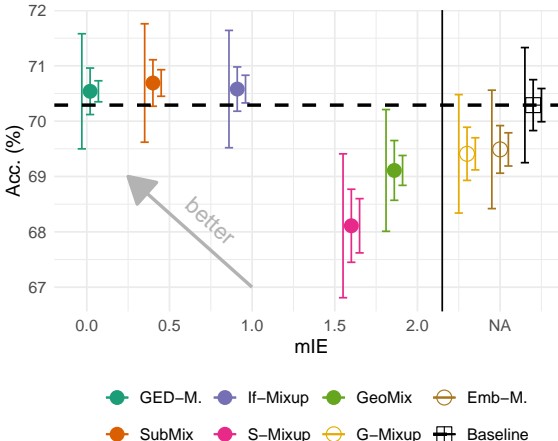

Figure 2: Mean interpolation error (mIE) and resulting test accuracy (%) along with standard errors under A1–A3 (left to right).

Table 5: Effect of mIE on empirical performance (test accuracy (%) $\pm$ standard errors (pp)) on IMDB-MULTI with GCN.

| Method | $\approx$ mIE | Acc. $\pm$ SE |
|---|---|---|
| GED-Mixup | 0.000 | $46.16 \pm 0.76$ |
| with | 0.250 | $44.98 \pm 1.02$ |
| injected | 0.500 | $44.67 \pm 0.83$ |
| interpolation | 0.625 | $44.04 \pm 0.82$ |
| errors | 0.750 | $41.33 \pm 1.12$ |
| | 0.875 | $39.09 \pm 1.36$ |
| | 1.000 | $35.51 \pm 1.92$ |
| Random guessing | | $\approx 33.33$ |

our results. As can be seen, all methods with high interpolation error as well as Emb-Mixup and G-Mixup (which do not perform pairwise interpolation of graphs) provided clearly inferior results compared to methods that interpolated better (GED-Mixup, SubMix, If-Mixup). This statement is statistically significant under all of our assumptions A1–A3 ($p = 5.57 \times 10^{-02}, 7.61 \times 10^{-06}, 1.40 \times 10^{-13}$ resp.). Tab. 5 reports results where we injected interpolation errors into GED-Mixup in a controlled fashion (see App. C.4 for details). We found that as the interpolation error grows, the empirical performance essentially deteriorated to random guessing. All of these findings provide evidence that bad interpolation properties can be detrimental for empirical performance.

Nevertheless, even optimal interpolation did not lead to statistically significant improvements (see also Tab. 2 and Tab. 3), i.e., good interpolation was not sufficient (see also the discussion in App. A.3).

**Detailed analysis.** To shed some light into the graphs produced by each of the methods, we visualize properties of the mixup graphs on the MUTAG dataset[7] for a choice of $\lambda = 0.8 \pm \varepsilon$ in Fig. 3, including the resulting mIE score. The plot represents each augmented pair $(G_1, G_2, G_M, \lambda)$ by a (slightly transparent) horizontal line, where the height correspond to the distance between the input graphs ($y = d(G_1, G_2)$) and the start- and endpoint to the distance of the mixup graph to each input (from $x_1 = d(G_1, G_M)$ to $x_2 = d(G_2, G_M)$. The blue targets indicate the points where $d(G_1, G_M) = \lambda d(G_1, G_2)$ and $d(G_2, G_M) = (1-\lambda)d(G_1, G_2)$, i.e., IC$_1$ and IC$_2$, resp., are satisfied. Ideally, all lines start and end at their target. The area marked in red shows particularly troublesome cases: if an endpoint falls into this area, the corresponding mixup graph has a larger distance to one of the inputs than the distance between the inputs themselves. In addition to not hitting the interpolation

---

[7]The conclusions for the MUTAG dataset are representative for the other datasets as well; see App. G.2.

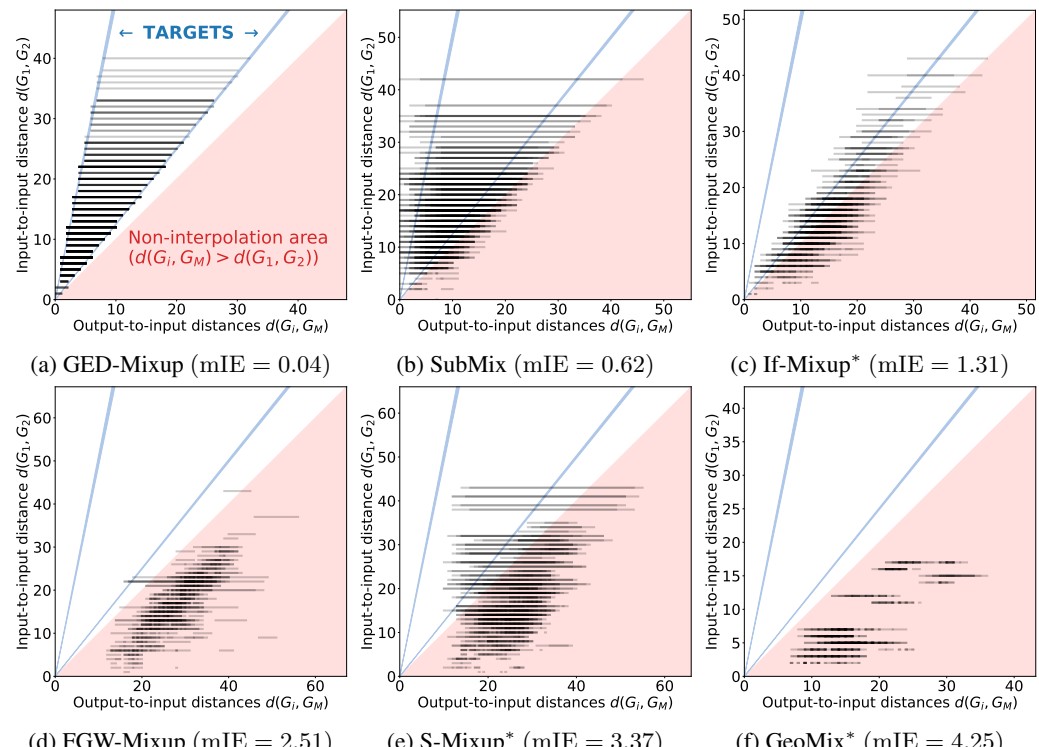

Figure 3: Visualization of mixup graphs produced on the MUTAG dataset. Each horizontal line corresponds to an input pair and its mixup graph and should ideally start and end at the blue targets.

targets, such graphs cannot even be interpreted as an interpolation between their inputs (corresponds to the non-green area in Fig. 1).

As expected, GED-Mixup interpolated well between inputs. For SubMix, which also interpolated well, we can see that the reason it slightly fell behind GED-Mixup interpolation is that it typically over- or undershot interpolation targets. If-Mixup, which fell behind considerably, produced results that roughly satisfied one of the targets ($IC_2$) but not the other one ($IC_1$). The mixup graphs produced by all other methods could generally not be treated as interpolations (red area is almost always touched).

**Related work.** Note that some prior work also performed structural analysis of mixup graphs to some extent. In particular, Zeng et al. (2024) evaluate structural plausibility of their GeoMix method using the Gromov-Wasserstein distance, i.e., within the Gromov-Wasserstein space rather than the input space. Their conclusions heavily depend on the suitability of that space, which our results call into question. Moreover, Ling et al. (2022) analyze their S-Mixup method using a variant of GED (which ignores distance between inputs, for example); our result indicate that S-Mixup does not exhibit good interpolation properties even though it optimizes that variant. Both works are limited in that they analyzed their respective proposed methods only, whereas our work provides a more holistic view.

## 6 CONCLUSION

We performed an independent evaluation of in a unified experimental setup following established evaluation protocols and systematically analyzed the mixup graphs produced by existing mixup methods from an interpolation perspective. While we do believe that mixup can be beneficial for graph classification tasks, our experimental study did not provide evidence for its efficacy. However, it also did not provide evidence to the contrary. We did find evidence that high interpolation errors lead to inferior results, though, which indicates that interpolation properties should be taken into account in subsequent works. Moreover, the question of how to select a suitable mixup graph (from a set of well-interpolating candidate graphs) is a promising direction for future work (see App. A.3).

## REPRODUCIBILITY STATEMENT

All of our work is reproducible. For this, we provided pseudo code for our analysis tool, GED-Mixup, both in the main text in Sec. 4.3 and in App. E with more details. Our experimental design for the interpolation analysis is described in Sec. 5, the experimental design for the empirical performance is summarized in Sec. 3 in the main text and described in detail in App. C.2. The source code required to reproduce our experiments is provided alongside this submission.

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

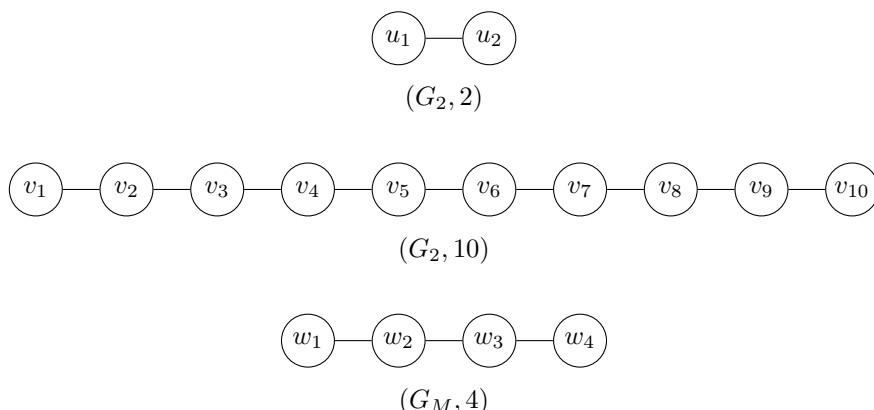

Figure 4: A toy dataset consisting of two path graphs $G_1$ and $G_2$. Each graph $G = (V, E)$ has the target label $|V|$. The mixup result $(G_M, 4)$ interpolates optimally between $(G_1, 2)$ and $(G_2, 10)$ for mixup ratio $\lambda = 1/4$.

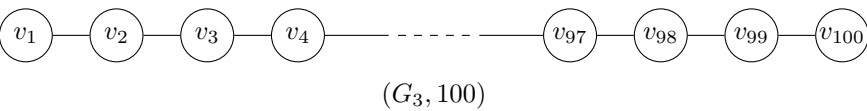

Figure 5: Graph $G_3$ with 100 nodes.

## A  THE PROMISE, THE SCOPE, AND THE CHALLENGES OF GRAPH MIXUP

In the section, we briefly discuss why graph mixup can be beneficial, how graph mixup differs from general data augmentation, and why graph mixup is challenging. We use handcrafted graph-level regression tasks throughout for accessibility; similar arguments apply to graph classification.

### A.1  THE PROMISE OF GRAPH MIXUP

Intuitively, the goal of graph mixup is to enhance training data with additional, useful training signals. Ideally, graph mixup produces correctly-labeled graphs from the data distribution, which are not yet part of the training data. The following example highlights that graph mixup can, indeed, achieve this goal.

Consider a regression task, where input graphs are path graphs and the task is to predict the number of nodes in the input graph. Suppose the training data consists of just two examples $(G_1, 2)$ and $(G_2, 10)$, see Fig. 4. Since the amount of training data is so small, overfitting is a serious concern.

When we apply GED-Mixup, which interpolates optimally, it is easy to see that all resulting mixup graphs are correctly-labeled (up to rounding; see Prop. 2) path graphs; see $(G_M, 4)$ in Fig. 4 for an example. In other words, graph mixup here provides useful training examples "for free".

This idealized example shows that graph mixup can be highly beneficial in principle. In our study, however, we did not find evidence for its benefits in any of the actual tasks we considered (see Sec. 3).

### A.2  THE SCOPE OF GRAPH MIXUP

One may wonder if optimal, or even decent, interpolation is necessary to obtain improved performance. We now argue that this is not the case.

Consider the setting of the preceding subsection, where the training set consists of two path-graph examples $(G_1, 2)$ and $(G_2, 10)$ and the task is to predict the number of nodes in the input graph. Any mixup method that interpolates optimally will produce graphs with 2–10 nodes; e.g., $(G_M, 4)$ in Fig. 4. Now consider graph $(G_3, 100)$ show in Fig. 5. Since this graph is substantially larger than

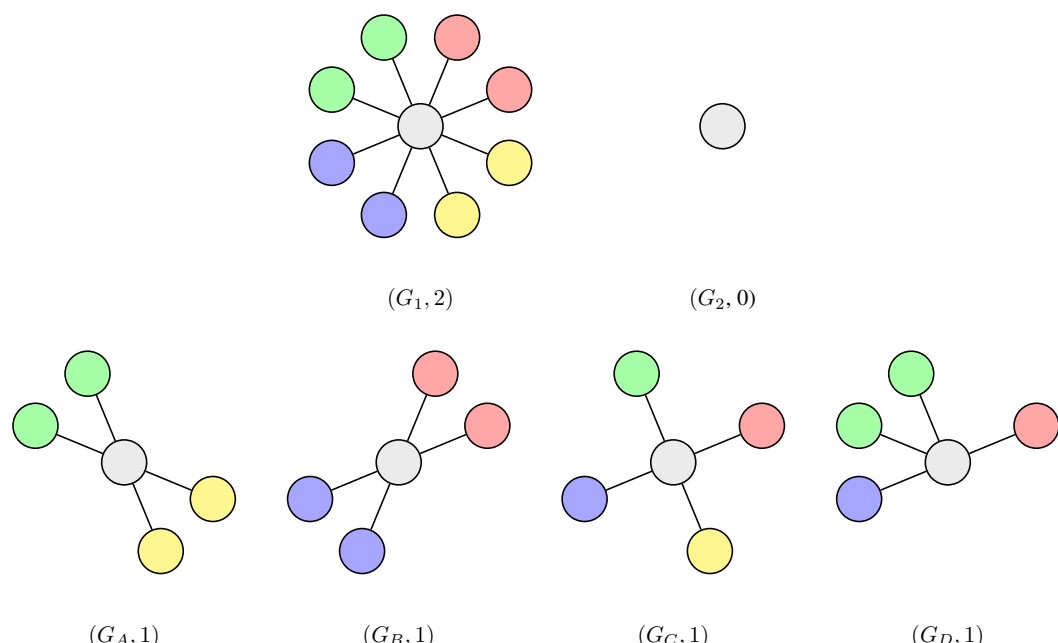

Figure 6: A dataset consisting of star graphs, where the task is to predict the number of blue nodes. Graphs $G_A, G_B, G_C$ and $G_D$ each interpolate optimally between inputs $G_1$ and $G_2$ for mixup ratio $\lambda = 1/2$.

both the training graphs and the mixup graphs, we generally do not expect that models learned on the training data (with or without mixup) generalize well in that they produce a good prediction for $G_3$.

Now consider a hypothetical "mixup" method that produces $(G_3, 100)$ when mixing up $(G_1, 2)$ and $(G_2, 10)$. This method will provide a useful training signal, but it does not interpolate well: $(G_3, 100)$ is far from the interpolation targets for any choice of $\lambda \in [0, 1]$. This pathological example shows that good interpolation properties are not necessary to provide a useful training signal.

However, we do not consider this hypothetical method a graph mixup method. In fact, the goal of all graph mixup methods (see Tab. 1 and the discussion in App. B) is to interpolate between input graphs, whereas the hypothetical method clearly does not (but instead is a different form of data augmentation). In this study, we exclusively restrict attention to graph mixup.

### A.3 THE CHALLENGES OF GRAPH MIXUP

In App. B, we argued that a main challenge in graph mixup is to obtain a good alignment of the input graphs. This is not the only challenge, however: even with an optimal alignment, selecting a suitable mixup graph can be challenging as well.

To make this more precise, consider a dataset consisting of star graphs as in Fig. 6 (top), where colors indicate different node features. We consider the regression problem of predicting the number of blue nodes in the input graph. Graphs $G_A$, $G_B$, $G_C$ and $G_D$ (bottom) are example mixup graphs that each interpolate optimally between the inputs $(G_1, 2)$ and $(G_2, 0)$. As can be seen, however, only $G_C$ and $G_D$ are correctly labeled (useful training signal), whereas $G_A$ and $G_B$ are not (misleading training signal).

The underlying challenge highlighted in this simple example is that *domain knowledge* may help to select suitable mixup graphs. Exploiting such domain knowledge is clearly beneficial. One of the primary advantages of graph mixup, however, is that it is domain independent. GED-Mixup, for example, simply chooses a random graph from all the mixup graphs that interpolate optimally. This choice is domain independent, but as the example shows, not ideal.

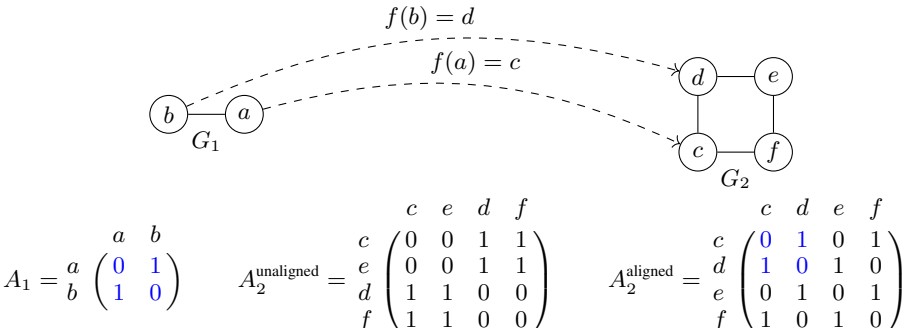

$$A_1 = \begin{array}{c} a \\ b \end{array} \begin{array}{cc} a & b \\ \begin{pmatrix} 0 & 1 \\ 1 & 0 \end{pmatrix} \end{array} \qquad A_2^{\text{unaligned}} = \begin{array}{c} c \\ e \\ d \\ f \end{array} \begin{array}{cccc} c & e & d & f \\ \begin{pmatrix} 0 & 0 & 1 & 1 \\ 0 & 0 & 1 & 1 \\ 1 & 1 & 0 & 0 \\ 1 & 1 & 0 & 0 \end{pmatrix} \end{array} \qquad A_2^{\text{aligned}} = \begin{array}{c} c \\ d \\ e \\ f \end{array} \begin{array}{cccc} c & d & e & f \\ \begin{pmatrix} 0 & 1 & 0 & 1 \\ 1 & 0 & 1 & 0 \\ 0 & 1 & 0 & 1 \\ 1 & 0 & 1 & 0 \end{pmatrix} \end{array}$$

Figure 7: The vertex mapping $f : V_1 \to V_2$ (dashed lines) maps the vertices $V_1$ of $G_1$ to the vertices $V_2$ of $G_2$. The vertex mapping $f$ is a full vertex mapping because $f(v_1)$ is well-defined for all $v_1 \in V_1$. The adjacency matrix $A_1$ corresponds to $G_1$, and both $A_2^{\text{unaligned}}$ and $A_2^{\text{aligned}}$ correspond to $G_2$. The matrix $A_1$ directly appears as a substructure in $A_2^{\text{aligned}}$ at the top left of the adjacency matrix, but not in $A_2^{\text{unaligned}}$.

We view the question of how to perform—in a domain-independent fashion—mixup given a suitable alignment as a promising direction for future work.

## B  PRELIMINARIES AND RELATED WORK

We first discuss vertex mappings and related concepts, and subsequently summarize existing mixup methods by relating them to vertex mappings.

### B.1  PRELIMINARIES: VERTEX MAPPINGS, EDIT SETS, GRAPH EDIT DISTANCE, ADJACENCY MATRICES

Consider two input graphs $G_1$ and $G_2$, with vertex sets $V_1$ and $V_2$ as well as edge sets $E_1$ and $E_2$, respectively. Without loss of generality, assume that $|V_1| \le |V_2|$. Two example graphs are shown in Fig. 7 (top). Note that the letters shown in the figure do not correspond to vertex labels (the graphs are unlabeled); we use them for expository reasons only.

A *vertex mapping* $f : V_1 \to V_2$ is an injective mapping from the vertices of $G_1$ to the vertices of $G_2$. Vertex mappings formalize the notion of an alignment of Sec. 2. One such mapping is shown in Fig. 7 (top).

Every vertex mapping $f$ induces an edit set, which is obtained by performing the edits required to obtain $G_2$ from $G_1$ by "transforming" each vertex $v_1 \in V_1$ (along with its neighborhood) to $f(v_1) \in V_2$ (likewise). For example, if $(v_1, u_1) \notin E_1$ but $(f(v_1), f(u_1)) \in E_2$, we include an insertion operation of edge $(v_1, u_1)$ to the edit set. The edit set induced by the vertex mapping shown in Fig. 7 is given by

$$\{ \text{insert vertex } e, \text{insert vertex } f, \text{insert edge } (b, e), \text{insert edge } (a, f), \text{insert edge } (e, f) \} .$$

The induced edit set can be computed in asymptotically linear time with respect to the number of nodes and edges (cf. Alg. 1 in (Chang et al., 2020)).

A vertex mapping is *optimal* if the size of its induced edit set is as small as possible. The GED $d(G_1, G_2)$ is given by the size of the edit sets of optimal vertex mappings. State-of-the-art algorithms for exact GED computation such as (Chang et al., 2023; 2020; Blumenthal et al., 2020) (sometimes implicitly) produce an optimal vertex mapping and hence a corresponding edit set as a by-product. We exploit this fact in our implementation of GED-Mixup.

For example, the computational framework used by Chang et al. (2020) and its improved version (Chang et al., 2023) starts by constructing a prefix-shared search tree, where each path from the root to a leaf represents a *full vertex mapping* $f$, and each path from the root to a non-leaf represents a *partial vertex mapping* $f_p$, in which $f_p(v_1)$ is undefined for some $v_1 \in V_1$. Each node is associated

with a *cost*, which constitutes a lower bound on (or, for leaves, the exact value of) the sizes of the induced edit sets obtained by all possible completions of the node's partial vertex mapping. The goal is to find a minimal-cost leaf, e.g., by using $A^*$ search. In order to obtain an efficient algorithm, lower bounds should (i) be as tight as possible so that pruning is effective and (ii) be efficiently computable.

We can alternatively represent vertex mappings by *vertex orderings*, i.e., an ordering of the vertices in $V_1$ and $V_2$, as well as by the corresponding *adjacency matrices*. For example, Fig. 7 shows adjacency matrix $A_1$ of $G_1$ for vertex ordering $(a, b)$. Likewise, $A_2^{\text{unaligned}}$ is the adjacency matrix of $G_2$ for vertex ordering $(c, e, d, f)$. The corresponding vertex mapping $f^{\text{unaligned}}$ is given by $f^{\text{unaligned}}(a) = c$ and $f^{\text{unaligned}}(b) = e$, i.e., the position of every vertex $v_1$ the ordering of $V_1$ matches the position of $f^{\text{unaligned}}(v_1)$ in the ordering of $V_2$. Vice versa, we can *align* the adjacency matrix of $G_2$ to the adjacency matrix $A_1$ with respect to a vertex mapping $f$. The corresponding adjacency matrix is shown as $A_2^{\text{aligned}}$.

Given two adjacency matrices $A_1$ and $A_2$, we can obtain an edit set by transforming $A_1$ to $A_2$ using (i) vertex insertions (inserting a zero row/column at the bottom/right of $A_1$), (ii) edge insertions (flip a 0 to a 1), (iii) edge deletions (flip a 1 to a 0), and (iv) vertex and edge relabelings (not shown).

We are now ready to describe existing graph mixup methods.

### B.2 METHODS FOR GRAPH MIXUP

We continue to use the setup and notation established in the previous section.

**If-Mixup (Guo and Mao, 2023).** If-Mixup uses adjacency matrices $A_1$ and $A_2$ obtained by a "default" vertex ordering of $V_1$ and $V_2$ present in the data. In more detail, it first pads $A_1$ (i.e., the smaller adjacency matrix) with zero rows and columns to obtain $A_1'$ (now of same size as $A_2$), and then performs linear mixup. The entries of the resulting adjacency matrix lie in $[0, 1]$ and can be interpreted as edge weights or edge existence probabilities (which are then fed into the GNN model as additional features). The result is heavily influenced by the vertex ordering present in the input data; e.g., for Fig. 7, it is less suitable when $A_2 = A_2^{\text{unaligned}}$ and more suitable when $A_2 = A_2^{\text{aligned}}$.

**S-Mixup (Ling et al., 2022).** S-Mixup can be seen as a variant of If-Mixup that aims to obtain a better alignment between the two adjacency matrices. Observe that the reordering operation involved in aligning an adjacency matrix $A_2$ to $A_1$ w.r.t. to vertex mapping $f$ can be expressed as $A_2^{\text{aligned}} = P A_2 P^\top$, where $P$ is a corresponding *alignment matrix* (which is a permutation matrix). S-Mixup uses a *soft alignment matrix* obtained from a Graph Matching Network (Li et al., 2019), which is trained on the available data. This soft alignment may not be optimal, but the hope is that it induces smaller edit sets than the "arbitrary" vertex mapping used by If-Mixup. Another difference is that S-Mixup, depending on the order of the two inputs, either produces a mixup graph with as many vertices as $G_1$ or a mixup graph with as many vertices as $G_2$.

**SubMix (Yoo et al., 2022).** SubMix is a mixup method inspired by CutMix (Yun et al., 2019). In CutMix, patches of images are cut and pasted between training examples, and the examples labels are mixed proportionally to the size of the patches. SubMix first samples a partial vertex ordering $V_1'$ of $V_1$ with a diffusion process (personalized page rank (Jeh and Widom, 2003)), likewise $V_2'$ of $V_2$. It then replaces the subgraph of $G_2$ induced by $V_2'$ by the subgraph of $G_1$ induced by $V_1'$. Since the partial vertex orderings are obtained from a random diffusion process, the quality of the obtained mixup graph depends on chance.

**G-Mixup (Han et al., 2022).** G-Mixup follows a different approach. It first computes graphons (Lovász and Szegedy, 2006), each corresponding to a class and summarizing all graphs of that class present in the training data. In more detail, a graphon is a symmetric function $W : [0, 1]^2 \to [0, 1]$ and can be interpreted as a "probabilistic adjacency matrix" of graphs of arbitrary sizes. Each probabilistic graph is represented by a set $V$ of vertex positions (each in $[0, 1]$), and the edge existence probability between vertices $u, v \in V$ is given by $W(u, v)$. To obtain a graphon, G-Mixup orders the vertices of the training graphs by degree, i.e., it implicitly uses a custom vertex ordering. To perform mixup, G-Mixup interpolates the graphons of two classes using linear

mixup, and then samples graphs from the resulting mixed graphon. Omeragic and Duranović (2023) reported reproducibility problems for this method.

**FGW-Mixup (Ma et al., 2023) and GeoMix (Zeng et al., 2024).** Both FGW-Mixup and GeoMix rely on the Gromov-Wasserstein distance (Vayer et al., 2020) from the theory of optimal transport (Villani, 2008). FGW-Mixup uses the Fused-Gromov-Wasserstein distance, which incorporates graph structure and features and aims to compute barycenters, wheres GeoMix uses the plain Gromov-Wasserstein distance and relies on geodesics Peyré et al. (2016). In both cases, mixup takes place in the Gromov-Wasserstein space. The optimal coupling between the input graphs obtained in the minimization of the (Fused-)Gromov-Wasserstein distance corresponds to a vertex mapping, i.e., these methods also aim to find a suitable mapping. Both FGW-Mixup and GeoMix are computationally expensive and hence make use of approximation algorithms.

**Embedding Mixup.** Mixup can also be performed in the embedding space of the GNN model (Wang et al., 2021), along the lines of manifold mixup (Verma et al., 2019). To do so, the current GNN model is used to compute embeddings of the two input graphs, and linear mixup is performed subsequently. Note that in contrast to the methods discussed before, embedding mixup does not generate a mixup graph. It also does not use an underlying vertex mapping, as the embeddings being interpolated are neural representations of entire input graphs.

**EPIC (Heo et al., 2024).** Edit Path Interpolation via Learnable Cost (EPIC) is based on the GED and associated edit paths between graphs, akin to our GED-Mixup baseline of Sec. 4.3. EPIC learns a cost model that aims to quantify the "importance" of specific edit operations from training data. GEDs are computed using this cost model and approximation algorithms. GED-Mixup is a simplified variant of EPIC in that it (i) uses unit edit costs (and thus does not require learning) and (ii) uses exact GED. We did not consider EPIC in our experimental study as there was no source code available.

## C EXPERIMENTAL DETAILS AND ADDITIONAL RESULTS

### C.1 CHOICE OF DATASETS AND DATASET STATISTICS

We chose the datasets for this experimental study based on the following criteria:

1. They are frequently used in the existing literature on graph mixup.
2. They cover a variety of domains:
   (a) Social networks (IMDB-BINARY, IMDB-MULTI)
   (b) Bioinformatics (ENZYMES, PROTEINS)
   (c) Molecular data (MUTAG, NCI1)
3. The experimental study is computationally feasible. In particular, methods such as FGW-Mixup and GeoMix have substantial computational cost, but we wanted to include these methods. Moreover, our goal was to use a unified, solid, and statistically sound experimental setup (see App. C.2). This further increases computational costs, as we performed extensive hyperparameter optimization (including of the no-mixup baseline), used cross-validation throughout to obtain more trustworthy performance estimates, and also analyzed interpolation properties (which requires GED computations).
4. The datasets are not too large. A key motivation of using graph mixup is to deal with data scarcity.

Summary statistics for all datasets are shown in Tab. 6. All datasets were obtained from TU-Dataset (Morris et al., 2020).

### C.2 EXPERIMENTAL SETUP (DETAILS)

This section outlines the details of the experimental setup that are summarized in Sec. 3 in the main text.

Table 6: Dataset Statistics

|  | **MUTAG** | **ENZYMES** | **IMDB-B** | **PROTEINS** | **IMDB-M** | **NCI1** |
|---|---|---|---|---|---|---|
| Domain | Molecules | Bioinf. | Social Networks | Bioinf. | Social Networks | Molecules |
| Graphs | 188 | 600 | 1000 | 1113 | 1500 | 4110 |
| Classes | 2 | 6 | 2 | 2 | 3 | 2 |
| Balanced? | No (65 vs. 125) | ✓ | ✓ | No (663 vs. 450) | ✓ | (✓)† |
| Avg. nodes | 17.93 | 32.63 | 19.77 | 39.06 | 13.00 | 29.87 |
| Avg. edges | 19.79 | 62.14 | 96.53 | 72.82 | 65.94 | 32.3 |
| Max. nodes | 28 | 126 | 136 | 620 | 89 | 111 |
| Max. edges | 66 | 298 | 2498 | 2098 | 2934 | 238 |
| Cat. node feat. | 7 | 3 | – | 3 | – | 37 |
| Cont. node feat. | – | 18 | – | 1 | – | – |
| Cat. edge feat. | 4 | – | – | – | – | – |
| Cont. edge feat. | – | – | – | – | – | – |
| References | (see below)[1] | (see below)[2] | (see below)[3] | (see below)[4] | (see below)[3] | (see below)[5] |

† NCI1 is approximately balanced (2053 vs. 2057).
[1] Debnath et al. (1991); Kriege and Mutzel (2012)
[2] Schomburg et al. (2004); Borgwardt et al. (2005)
[3] Yanardag and Vishwanathan (2015)
[4] Borgwardt et al. (2005); Dobson and Doig (2003)
[5] Wale et al. (2008); Shervashidze et al. (2011) and https://pubchem.ncbi.nlm.nih.gov/ (Pub-Chem)

**Models.** We considered the GCN (Kipf and Welling, 2017) and GIN (Xu et al., 2018) models in our structure. Both are simple, commonly used models and have sufficient representational capacity for the tasks we consider here. I.e., they did overfit in our experiments, a problem that mixup aims to alleviate.

**Mixup methods.** We considered all methods discussed in Sec. 2 for which open-source implementations were available, i.e., If-Mixup (Guo and Mao, 2023), S-Mixup (Ling et al., 2022), SubMix (Yoo et al., 2022), G-Mixup (Han et al., 2022), FGW-Mixup (Ma et al., 2023), GeoMix (Zeng et al., 2024), and Emb-Mixup (Wang et al., 2021).

**Training.** Given a labeled training set and a choice of hyperparameters, we trained each model in a common training pipeline based on PyTorch Geometric (Fey and Lenssen, 2019) and Optuna (Akiba et al., 2019), using cross entropy loss. To perform mixup, we used the original implementations from prior work as well as our implementation of GED-Mixup. In each epoch, we generated a fixed number of mixup graphs (a hyperparameter) and added them to the training data for this epoch. We followed the common approach of producing mixup graphs by mixing two randomly chosen graphs/labels from the training data, using mixup ratio $\lambda \sim \mathrm{Beta}(\alpha, \alpha)$ (for hyperparameter $\alpha \in \mathbb{R}^+$).

**Methodology.** We followed Errica et al. (2019), who describe key criteria for evaluating methods for graph classification tasks. This includes (i) nested cross-validation (CV) for model selection (inner CV for hyperparameter search) and model assessment (outer CV for test), (ii) repeat model assessment multiple times to account for training randomness (e.g., model initialization), and (iii) publish source code and ensure reproducibility. These criteria are well-established in machine learning (Cawley and Talbot, 2010; Bengio, 2012; Bengio and Grandvalet, 2004) and particularly important for graph classification tasks due to small dataset sizes and a lack of predefined train-test splits.

We emphasize these points because we found that prior studies often did not fully adhere to such evaluation standards. For instance, some studies (Guo and Mao, 2023; Yoo et al., 2022; Wang et al., 2021) followed the evaluation protocol of Xu et al. (2018), despite its problematic use of validation rather than test performance (Errica et al., 2019). Other studies (Ling et al., 2022; Han et al., 2022) used holdout validation instead of a cross-validation or use cross-validation only for model selection but not for model assessment (Ma et al., 2023). While most methods provide source code of the proposed method (except for (Guo and Mao, 2023; Heo et al., 2024)), source code for other key aspects such as model selection is missing. Adding to these points, statistical significance was rarely assessed so that it is not clear whether reported improvements are real. As we will see, our study

answers this negatively, i.e., most methods failed to produce statistically significant improvements under a rigorous evaluation.

**Hyperparameters.** We used training hyperparameters (such as the learning rate or optimizer), model architecture hyperparameters (such as the number of layers, hidden dimensionalities, or dropout probability), and mixup hyperparameters (such as the number of added mixup graphs or sampling distribution of the mixup ratio $\lambda$). Tab. 7 summarizes our hyperparameters and search space. For GCN/GIN, we followed You et al. (2020). For specific mixup methods, we used the hyperparameter values or search spaces suggested in the code or the publication. Descriptions and rationales are given below the table.

**Tuning.** Given a training and a validation split, we sampled 10 hyperparameter configurations randomly and subsequently $n$ configurations using Bayesian optimization with TPE (Bergstra et al., 2011). Model selection was performed by mean validation accuracy (over the inner CV folds). We first tuned strong baseline models without mixup in this fashion (using $n = 90$), and subsequently tuned just the mixup hyperparameters (separately for each mixup method, using $n = 10$) in the same way. This ensures fairness, as all mixup methods used the same, well-performing model architecture.

**Metrics.** We used 5-fold nested cross-validation throughout, three repeated training runs for model assessment, and report mean classification accuracy on the test splits as a metric. We also report standard errors and statistical significance compared to the no-mixup baseline using Welch's two-sided $t$-test (Welch, 1947) with a significance level of $0.05$. We did not correct for multiple testing.

**Hardware and computational cost.** We required approximately 144 GPU hours per model and dataset to determine the GNN hyperparameters and performance. We required approximately 96 GPU hours to determine hyperparameters and performance for each mixup method and dataset. We used NVIDIA RTX 2080Ti and NVIDIA RTX A6000 GPUs supported by various generations of either Intel Xeon CPUs (such as E2640 v2, E5-2640 v3, E5-2698 v4, and Silver 4114) or various generations of AMD EPYC CPUs (such as 7413, 9474F, and 7713P).

**Software.** Our experimental pipeline is implemented with PyTorch Geometric (Fey and Lenssen, 2019) (MIT License) and Optuna (Akiba et al., 2019) (MIT License). We used the original mixup implementations whenever available:

- Emb-Mixup: (own implementation)
- FGW-Mixup: https://github.com/ArthurLeoM/FGWMixup (no license)
- GED-Mixup: (own implementation)
- GeoMix: https://github.com/zhichenz98/GeoMix-ICML24 (no license)
- G-Mixup: https://github.com/ahxt/g-mixup (no license)
- If-Mixup: (own implementation)
- SubMix: https://github.com/snudatalab/GraphAug (custom license available under the specified URL)
- S-Mixup: https://github.com/divelab/DIG (GPL-3.0 license)
- GED and edit set computation: AStar-BMao (Chang et al., 2023) available from GitHub https://github.com/LijunChang/Graph_Edit_Distance (MIT License). We modified the code to additionally provide vertex mappings.

C.3 COMPUTATIONAL COST OF GRAPH MIXUP

We report on the computational cost of mixup in this section. We found that, when suitably optimized, graph mixup was generally feasible but often added substantial runtime overhead. Moreover, GED-Mixup method was feasible for its intended purpose of analysis, even with the exact GED computations that we used.

**Experimental setup and results.** When training on datasets NCI1 and IMDB-MULTI, we maintained runtime statistics (wall clock time) per epoch. We report three values per method: (i) the per-epoch cost without mixup (baseline), (ii) the per-epoch cost with a naive application of mixup (naive), and (iii) the amortized per-epoch cost with an optimized application of mixup (optimized). In our experiments, we used option (iii) whenever applicable. Tab. 8 contains our results.

Table 7: Hyperparameter search space

|  | **Hyperparameter** | **Search space** |
| --- | --- | --- |
| Training | Optimizer | Adam (2015), SGD (1951) |
|  | Learning rate | $10^{-5}$ to $10^{-1}$ (log scale) |
|  | Batch size | $\{8, 16, 32, 64, 128, 256\}$ |
|  | Dropout probability (2014) | $[0, 0.5]$ |
|  | Early stopping[1] | $\{1, \ldots, 1000\}$ |
| GCN/GIN | No. pre-processing layers | $1, 2, 3$ |
|  | No. convolutional layers | $2, 4, \ldots, 8$ |
|  | No. post-processing layers | $1, 2, 3$ |
|  | Embedding size | 32 to 256 (log scale) |
|  | Readout | Mean, Max, Sum |
|  | Normalization | None, Batch norm (2015) |
| Mixup[*] | Augmentation ratio[2] | $[0.2, 2.0]$ |
|  | Keep original graphs?[3] | Yes or no |
|  | Mixup ratio parameter $\alpha$[4] | $\{0.1, 0.3, 0.5, 1.0, 5.0\}$ |
| FGW-Mixup | FGW-alpha[5] | $\{0.05, 0.5, 0.95\}$ |
|  | $\rho$[6] | $\{0.1, 1, 10\}$ |
| S-Mixup | GMNet batch size[7] | $\{8, 64, 128\}$ |
|  | No. GMNet layers[7] | $\{4, 5, 6\}$ |
| SubMix | Subgraph size[8] | $\{0.2, 0.4, 0.6\}$ |

[*] Used by all mixup methods unless stated otherwise.

[1] We determine the number of "early-stopping" epochs that leads to the best validation result, and use it when we retrain on the entire four folds for testing on the remaining fold. We do this so that test data is not used for early stopping and hence not leaked.

[2] Fraction of generated mixup graphs per epoch w.r.t. the size of the training data. Here we force each mixup method to add a non-trivial fraction (20%) of mixup graphs during training. We do this since we are primarily interested in whether mixup is effective and not in hyperparameter choices that do not actually add mixup graphs.

[3] Whether to include non-mixup graphs during training.

[4] Mixup ratio $\lambda$ is sampled from $\mathrm{Beta}(\alpha, \alpha)$. Does not apply to SubMix.

[5] Trade-off parameter between Gromov-Wasserstein cost on graph structure and Wasserstein cost on node features (cf. (Ma et al., 2023)).

[6] Step size hyperparameter in mirror descent (cf. (Ma et al., 2023)).

[7] Hyperparameter of the graph matching network (cf. (Li et al., 2019)).

[8] Relative size of the selected subgraphs (cf. (Yoo et al., 2022)).

Table 8: Average runtime per method and epoch on IMDB-MULTI and NCI1 (in seconds).

| Method | Baseline | Naive | Optimized |
|---|---|---|---|
| Baseline | 0.71 | – | – |
| Emb-Mixup | 0.71 | 0.55 | – |
| G-Mixup | 0.71 | 16.43 | – |
| GeoMix | 0.71 | 16,709.86 | 1.73 |
| If-Mixup | 0.71 | 1.57 | 0.99 |
| S-Mixup | 0.71 | 9.34 | – |
| SubMix | 0.71 | 7.23 | 1.07 |
| GED-Mixup | 0.71 | 4,813.62 | 1.06 |

**Optimizing graph mixup.** In more detail, (ii) naive generates the required mixup graphs individually for every epoch. As can be seen below, this can add substantial overhead, e.g., for GeoMix and GED-Mixup. Approach (iii), in contrast, pre-computes a large number of mixup graphs upfront and samples from these graphs during every epoch; we report the pre-computation cost amortized over all training epochs. This optimized approach is not novel and has been used in prior work (e.g., Han et al. (2022); Zeng et al. (2024)). The optimized approach is applicable, whenever the mixup is not learned/does not depend on the training process (i.e., for FGW-Mixup, GeoMix, If-Mixup, SubMix, GED-Mixup).

Since the optimized approach (iii) is not applicable to G-Mixup or S-Mixup, these methods had the highest computational cost. GeoMix, even when using the optimized approach (iii), required more time than GED-Mixup. Most mixup methods with the exception of Emb-Mixup added a substantial computational overhead over the no-mixup baseline.

**Limitations.** These experiments were conducted on a compute cluster with different kinds of GPUs and CPUs (see App. C.2), so that results are somewhat noisy.

### C.4 CONTROLLED INJECTION OF INTERPOLATION ERRORS (DETAILS)

Tab. 5 reports results where we injected interpolation errors into GED-Mixup in a controlled fashion (see App. C.4 for details). We describe the details of this experiment in what follows.

**Experimental setup.** We considered one model (GCN) and one dataset (IMDB-MULTI) using the respective tuned hyperparameters. We focused on GED-Mixup in this experiment, as it allowed us to precisely control the interpolation error (i.e., mIE).

To produce mixup graphs, we proceeded as in GED-Mixup but used a different mixup ratio for input graphs and input labels. Effectively, this moves the mixup graph closer to a randomly chosen input graph than desired, and thus immediately violates the two interpolation targets in a controlled fashion.

In more detail, given a desired interpolation error $\varepsilon$ as input ($\approx$ mIE in Tab. 5), instead of using interpolation ratio $\lambda$, we mix with $\lambda \pm \varepsilon$ (clipped to $[0, 1]$). Since GED-Mixup interpolates optimally (recall Prop. 2), this leads to an expected mIE of $\varepsilon$.

**Limitations.** We used one dataset and one model only. Moreover, all mixup graphs remained on an optimal edit path between the input graphs. This may be seen as a somewhat optimistic setting, yet it sufficed to observe degradation in empirical performance. Moreover, there are many ways to "deviate" from the interpolation targets; this is perhaps the simplest one. We used it here because it is both interpretable and feasible to implement as well as run.

### C.5 LABEL NOISE

A reviewer of this paper pointed out that one of the application areas of mixup is to mitigate the impact of label noise. We conducted an small experiment to investigate to what extent graph mixup may help here. In line with our other results, we found that none of the mixup methods performed better or worse (with statistical significance) than when using no mixup.

**Experimental setup and results.** We focused on GIN and IMDB-BINARY, and injected noise by flipping labels randomly with label flipping probabilities of $p \in \{0, 0.125, 0.25, 0.5\}$ during

Table 9: Label noise experiment with GIN on IMDB-BINARY (test acc. (%) $\pm$ standard error (pp)) for label flipping probabilities $p \in \{0, 0.125, 0.25, 0.5\}$. Statistically significant differences over the no-mixup baseline are marked bold (there are none; $p$-values in parentheses).

| Method | 0 | 0.125 | 0.25 | 0.5 |
|---|---|---|---|---|
| Baseline | $70.77 \pm 0.53$ | $69.43 \pm 1.08$ | $66.80 \pm 1.51$ | $48.27 \pm 2.40$ |
| Emb-M. | $67.77 \pm 1.89$ (0.15) | $69.07 \pm 1.75$ (0.86) | $68.53 \pm 0.93$ (0.34) | $48.33 \pm 1.64$ (0.98) |
| G-Mixup | $65.90 \pm 2.38$ (0.06) | $67.27 \pm 2.05$ (0.36) | $68.07 \pm 1.06$ (0.50) | $45.33 \pm 2.21$ (0.38) |
| GeoMix | $70.53 \pm 0.60$ (0.77) | $68.17 \pm 0.90$ (0.38) | $65.20 \pm 1.54$ (0.46) | $47.40 \pm 1.32$ (0.75) |
| If-Mixup | $69.30 \pm 0.72$ (0.11) | $67.90 \pm 1.67$ (0.45) | $67.30 \pm 1.54$ (0.82) | $47.27 \pm 1.46$ (0.73) |
| S-Mixup | $69.29 \pm 2.16$ (0.52) | $68.33 \pm 1.63$ (0.58) | $66.87 \pm 0.93$ (0.97) | $48.60 \pm 1.53$ (0.91) |
| SubMix | $70.40 \pm 0.45$ (0.60) | $69.07 \pm 1.32$ (0.83) | $65.73 \pm 1.88$ (0.66) | $47.40 \pm 1.01$ (0.74) |
| GED-M. | $70.40 \pm 0.76$ (0.70) | $65.67 \pm 2.26$ (0.15) | $63.20 \pm 2.32$ (0.21) | $47.23 \pm 1.58$ (0.72) |

Table 10: $p$-values associated to the results presented in Tab. 2. Values below $0.05$ are highlighted in **bold** (significantly worse than baseline).

| Method | MUTAG | ENZYMES | IMDB-BINARY | PROTEINS | IMDB-MULTI | NCI1 |
|---|---|---|---|---|---|---|
| | | | GCN | | | |
| Emb-M. | 0.58 | 0.34 | 0.31 | 0.64 | **0.02** | 0.06 |
| G-Mixup | 0.07 | 0.08 | 0.69 | 0.47 | 0.60 | 0.48 |
| GeoMix | 0.48 | 0.40 | **0.02** | 0.96 | **0.02** | 0.34 |
| If-Mixup | 0.20 | 0.75 | 0.86 | 0.34 | 0.38 | 0.12 |
| S-Mixup | 0.46 | 0.36 | 0.52 | 0.90 | **0.01** | 0.71 |
| SubMix | 0.53 | 0.66 | 0.71 | 0.78 | 0.36 | 0.11 |
| GED-M. | 0.21 | 0.81 | 0.92 | 0.63 | 0.87 | 0.06 |
| | | | GIN | | | |
| Emb-M. | 0.20 | 0.76 | 0.15 | 0.76 | 0.88 | 0.61 |
| G-Mixup | 0.09 | 0.44 | 0.06 | 0.79 | 0.48 | 0.26 |
| GeoMix | 0.33 | 0.42 | 0.77 | 0.92 | 0.54 | 0.88 |
| If-Mixup | 0.87 | 0.87 | 0.11 | 0.97 | 0.83 | 0.87 |
| S-Mixup | **0.04** | 0.42 | 0.52 | 0.08 | 0.18 | 0.10 |
| SubMix | 0.88 | 0.82 | 0.60 | 0.79 | 0.87 | 0.37 |
| GED-M. | 0.42 | 0.55 | 0.70 | 0.74 | 0.85 | 0.55 |

training (but not during testing, of course). To keep costs controlled, we re-used the clean-data hyperparameters from our study. Tab. 9 shows our results.

**Limitations.** We investigated only a single model, only a single dataset, and did not conduct separate hyperparameter optimization. Nevertheless, the results are not promising.

## C.6 P-VALUES FOR TAB. 2

We report $p$-values for our empirical results of Tab. 2 below in Tab. 10.

## D POOLED ANALYSIS

We estimate standard errors under assumptions (A1)–(A3) discussed in Sec. 3. We start with (A1) and subsequently discuss (A2)–(A3).

**Observations.** Consider an arbitrary but fixed mixup method. In our evaluation, we considered

- Two model architectures $\mathcal{M} = \{\,\text{GCN}, \text{GIN}\,\}$,

- Six data distributions $\mathcal{D} = \{\,\text{MUTAG}, \text{ENZYMES}, \ldots, \text{NCI1}\,\}$,

- Five folds $\mathcal{F}_d = \{\, f_1^d, \ldots, f_5^d \,\}$ per data distribution $d \in \mathcal{D}$, each consisting of a training and a test split, and

- Three runs $\mathcal{R}_{mdf} = \{\, r_1^{mdf}, r_2^{mdf}, r_3^{mdf} \,\}$ per model architecture $m \in \mathcal{M}$, data distribution $d \in \mathcal{D}$, and fold $f \in \mathcal{F}_d$.

We then observe the accuracies $A_{mdfr} \in [0, 1]$ of run $r$ on fold $f$ for data distribution $d$ and model architecture $m$.

**Assumption (A1, standard).** In what follows, we use capital letters to refer to random variables (e.g., $D$ for a random data distribution) and small as well as calligraphic letters to refer to concrete realizations (e.g., $d$ for a concrete realization of $D$ and $\mathcal{D}$ for multiple such realizations). Under (A1), we assume that

(A1-M) model architectures are drawn from a distribution $p(M)$ of "real-world" model architectures,

(A1-D) data distributions are drawn from a distribution $p(D)$ of "real-world" data distributions,

(A1-F) folds $F$ are drawn from a fold distribution $p(F_D)$ obtained by sampling each element of each of the training/test splits independently from data distribution $D$,

(A1-R) runs $R$ are drawn from a run distribution $p(R_{MDF})$ determined by the random choices made during training (such as randomness in initialization, batch construction, or mixup).

These assumptions allow us to estimate the impact of mixup beyond the concrete datasets and model architectures used in this study.

Note that we make a key simplifying assumption here: we treat each observation $A_{mdfr}$ as an independent realization of $A_{MDFR}$. By doing so, we ignore that in our implementation, (i) all 3 runs use the same fold and hyperparameters, and (ii) all 5 folds are obtained by cross-valuation and from a single dataset. This may lead to an underestimation of variance (Bengio and Grandvalet, 2004). We proceed this way to keep the cost of the experimental study controlled, and we need this simplifying assumption to make analysis feasible.

**Estimators.** We estimate the expected accuracy $A_{mdf} = \mathbb{E}_{R \sim p(R_{mdf})}[A_{mdfR}]$ of model architecture $m$ on fold $f$ for data distribution $d$ by the sample mean, i.e.,

$$\hat{A}_{mdf} = \frac{1}{|\mathcal{R}_{mdf}|} \sum_{r \in \mathcal{R}_{mdf}} A_{mdfr}.$$

Likewise, the expected accuracy $A_{md} = \mathbb{E}_{F \sim p(F_d)}[A_{mdF}]$ of model architecture $m$ on data distribution $d$ is estimated as

$$\hat{A}_{md} = \frac{1}{|\mathcal{F}_d|} \sum_{f \in \mathcal{F}_d} \hat{A}_{mdf}.$$

This estimate is shown in Tab. 2 for various mixup methods. Finally, the estimates for the expected performance $A_m \sim \mathbb{E}_{D \sim p(D)}[A_{mD}]$ of model architecture $m$ and expected overall performance $A = \mathbb{E}_{M \sim p(M)}[A_M]$ are obtained similarly, i.e.,

$$\hat{A}_m = \frac{1}{|\mathcal{D}|} \sum_{d \in \mathcal{D}} \hat{A}_{md}$$

$$\hat{A} = \frac{1}{|\mathcal{M}|} \sum_{m \in \mathcal{M}} \hat{A}_m.$$

The latter estimate is shown in Tab. 3 for various mixup methods.

**Standard errors.** We now derive the standard errors of each of the above estimators under (A1). First, we estimate the variance $\sigma_{mdf}^2 = \mathrm{Var}_{R \sim p(R_{mdf})}[A_{mdfR}]$ by the sample variance, i.e.,

$$\hat{\sigma}_{mdf}^2 = \frac{1}{|\mathcal{R}_{mdf}| - 1} \sum_{r \in \mathcal{R}_{mdf}} (A_{mdfr} - \hat{A}_{mdf})^2.$$

For the variance $\sigma_{md}^2 = \mathrm{Var}_{F \sim p(F_d)}[A_{mdF}]$, we use the law of total variance to obtain

$$\sigma_{md}^2 = \underbrace{\mathbb{E}_{F \sim p(F_d)}[\mathrm{Var}(A_{mdF}|F)]}_{\text{``within-fold''}} + \underbrace{\mathrm{Var}_{F \sim p(F_d)}(\mathbb{E}[A_{mdF}|F])}_{\text{``between-folds''}}.$$

We estimate the first part by the mean within-fold variance estimate and the second part by the sample variance of estimates across folds:

$$\hat{\sigma}_{md}^2 = \underbrace{\frac{1}{|\mathcal{F}|} \sum_{f \in \mathcal{F}_d} \hat{\sigma}_{mdf}^2}_{\text{``within-fold''}} + \underbrace{\frac{1}{|\mathcal{F}| - 1} \sum_{f \in \mathcal{F}_d} (\hat{A}_{mdf} - \hat{A}_{md})^2}_{\text{``between-folds''}}.$$

Using the simplifying assumption of independent observations discussed above, we estimate the standard error of $\hat{A}_{md}$ by

$$\widehat{\mathrm{SE}}(\hat{A}_{md}) = \sqrt{\frac{\hat{\sigma}_{md}^2}{n_{md}}},$$

where $n_{md} = \sum_{f \in \mathcal{F}_d} \sum_{r \in \mathcal{R}_{mdf}} 1 = 15$. This standard error is shown in Tab. 2.

Using the same approach, we obtain an estimate of the variance $\sigma_m^2 = \mathrm{Var}_{D \sim p(D)}[A_{mD}]$ as

$$\hat{\sigma}_m^2 = \underbrace{\frac{1}{|\mathcal{D}|} \sum_{d \in \mathcal{D}} \hat{\sigma}_{md}^2}_{\text{``within-dataset''}} + \underbrace{\frac{1}{|\mathcal{D}| - 1} \sum_{d \in \mathcal{D}} (\hat{A}_{md} - \hat{A}_m)^2}_{\text{``between-datasets''}}$$

and of variance $\sigma^2 = \mathrm{Var}_{M \sim p(M)}[A_M]$ as

$$\hat{\sigma}^2 = \underbrace{\frac{1}{|\mathcal{M}|} \sum_{m \in \mathcal{M}} \hat{\sigma}_m^2}_{\text{``within-model''}} + \underbrace{\frac{1}{|\mathcal{M}| - 1} \sum_{m \in \mathcal{M}} (\hat{A}_m - \hat{A})^2}_{\text{``between-models''}}.$$

We estimate the standard error of $\hat{A}$ by

$$\widehat{\mathrm{SE}}(\hat{A}) = \sqrt{\frac{\hat{\sigma}^2}{n}},$$

where $n = \sum_{m \in \mathcal{M}} \sum_{d \in \mathcal{D}} \sum_{f \in \mathcal{F}_d} \sum_{r \in \mathcal{R}_{mdf}} 1 = 150$. This standard error is shown in the (A1) column of Tab. 3.

**Assumption (A2, fixed distribution).** For (A2), we make the stronger assumption that $\mathcal{M}$ and $\mathcal{D}$ are not samples (from a distribution of model architectures and a distribution of data distributions, respectively) but the entire population of interest. Then $A_m = \frac{1}{|\mathcal{D}|} \sum_d A_{md}$ and $A = \frac{1}{|\mathcal{M}|} \sum_{m \in \mathcal{M}} A_m$ become means (instead of being expected values). As a consequence, the "between-datasets" and the "between-models" terms in our estimate of $\hat{\sigma}_m^2$ and $\hat{\sigma}^2$, respectively, vanish. The corresponding standard errors are shown in the (A2) column of Tab. 3.

**Assumption (A3, fixed data).** For (A3), we make assumption (A2) plus the additional assumption that the folds in $\mathcal{F}_d$ are not samples (from the data distribution $d$) but the entire population of interest. Then $A_{md} = \frac{1}{|\mathcal{F}_d|} \sum_{f \in \mathcal{F}_d} A_{mdf}$ becomes a mean (instead of being an expected value). As a consequence, the "between-folds" term in our estimate of $\hat{\sigma}_{md}^2$ vanishes as well. The corresponding standard errors are shown in the (A3) column of Tab. 3.

---

**Algorithm 2** GED-Mixup of Alg. 1 in more detail

---

**Require:** Graphs $G_1, G_2$; mixup ratio $\lambda \in [0, 1]$
**Ensure:** Mixup graph $G_M$
1: $F^* \leftarrow$ a minimal edit set between $G_1$ and $G_2$, obtained by AStar-BMao (Chang et al., 2023)
2: **repeat**
3:      $G_M \leftarrow$ ADMISSIBLEMIXUP$(G_1, F^*, \text{round}(\lambda |F|))$
4: **until** $G_M$ is connected **or** $G_1$ is not connected **or** $G_2$ is not connected
5: **return** $G_M$
6:
7: **function** ADMISSIBLEMIXUP$(G_1, F, n)$
8:      $G_M \leftarrow G_1$
9:      **for** $i \leftarrow 1, \ldots, n$ **do**
10:         $f \leftarrow$ sample an edit operation from $F$ that is admissible on $G_M$          ▷ App. E
11:         $G_M \leftarrow \text{apply}(G_M, f)$
12:         $F \leftarrow F \setminus \{ f \}$
13:      **end for**
14:      **return** $G_M$
15: **end function**

---

# E    DETAILS OF GED-MIXUP

Alg. 2 shows a slightly more detailed version of Alg. 1. In our implementation, we use AStar-BMao (Chang et al., 2023), an algorithm for GED computation, and modified it to also output a corresponding edit set $F$.

In contrast to Alg. 1, Alg. 2 does not first sample an edit path and subsequently generate a mixup graph. Instead, it generates $G_M$ directly.

Akin to the discussion in Sec. 4.3, we consider an edit path as *valid* if its operations (when applied in order) each satisfy: (i) an edge can only be inserted when its source and target vertex are present, (ii) a vertex can only be removed when it does not have an incident edge, (iii) label substitutions are only possible for vertices/edges present in the graph, and (iv) when both $G_1$ and $G_2$ are connected, so is $G_M$. We check for conditions (i)—(iii) as we go: ADMISSIBLEMIXUP repeatedly samples a not-yet-used and *admissible* operation—i.e., an operation satisfying (i)—(iii)—from $F$ and returns the resulting mixup graph. If the mixup graph also satisfies (iv), we output it, otherwise we repeat the sampling process.[8]

Note that our approach to obtain an edit path is rather naive (see App. A.3): The obtained path is "random" and largely ignores locality of edit operations. Since we view GED-Mixup as a baseline, however, we did not explore this further.

# F    PROOFS

## F.1    PROOF OF PROP. 1

**Proposition 1.** *For any distance metric $d(a, b) = \|a - b\|$ induced by a norm $\|\cdot\|$ on the input/label vector space (over $\mathbb{R}$), linear mixup of Eq. (1) satisfies IC for inputs/labels and we have*

$$\text{AIE}_d(x_M) = \text{IE}_d(x_M) = 0 \quad / \quad \text{AIE}_d(y_M) = \text{IE}_d(y_M) = 0.$$

*Proof.* Consider inputs $x_1$ and $x_2$, any $\lambda \in [0, 1]$, and any norm $\|\cdot\|$ on the input space. Linear mixup produces the result

$$x_M = x_1 + \lambda(x_2 - x_1).$$

---

[8]Assume $G_1$ and $G_2$ are connected. For some edit sets, every admissible edit path of length $\text{round}(\lambda |F|)$ produces a mixup graph that is not connected. This problem can be fixed by allowing one more or one less edit operation. In our implementation, we use a simpler approach and abort generation if we do not obtain a valid mixup graph after 10 repetitions. This is not a problem in practice, as we can simply sample a new value for $\lambda$ or a new set of graphs to mixup.

We have

$$
\begin{aligned}
d(x_1, x_M) &= \|x_1 - x_M\| \\
&= \|x_1 - (x_1 + \lambda(x_2 - x_1))\| \\
&= \|-\lambda(x_2 - x_1)\| \\
&= \lambda \|(x_2 - x_1)\| \\
&= \lambda \cdot d(x_1, x_2),
\end{aligned}
$$

which proves $\mathrm{IC}_1$. The same arguments can be made for $\mathrm{IC}_2$ and as well as $y_M$ so that all interpolation errors are zero as claimed. $\qquad\square$

### F.2  PROOF OF PROP. 2

**Proposition 2.** *GED-Mixup (Alg. 1) interpolates optimally w.r.t. $d_{GED}$, and it holds*

$$
\mathrm{AIE}(G_M) = 2 \cdot |\mathrm{round}(\lambda \cdot d_{GED}(G_1, G_2)) - \lambda \cdot d_{GED}(G_1, G_2)| \leq 1.
$$

*Proof.* Consider inputs $G_1$ and $G_2$, let $P$ be the edit path chosen by Alg. 1. By construction, we have

$$
\mathrm{apply}(G_1, P) \cong G_2 \quad \text{and} \quad d_{12} \overset{\text{def}}{=} d_{\mathrm{GED}}(G_1, G_2) = |P|.
$$

Denote by

$$
d_{M1} = \lambda \cdot |P| = \lambda \cdot d_{12}
$$

the interpolation target ($\mathrm{IC}_1$) and by $P_\lambda$ the first $\mathrm{round}(d_{M1})$ edit operations in $P$. Alg. 1 produces

$$
G_M = \mathrm{apply}(G_1, P_\lambda)
$$

and it holds that

$$
d(G_M, G_1) = \mathrm{round}(d_{M1})
$$

To see this, first observe that $P_\lambda$ contains $\mathrm{round}(d_{M1})$ edit operations so that $d(G_1, G_M) \leq \mathrm{round}(d_{M1})$. Now suppose for contradiction that $d(G_1, G_M) < \mathrm{round}(d_{M1})$ and let $P'_\lambda$ be a corresponding edit path. By replacing $P_\lambda$ by $P'_\lambda$ in $P$, we obtain an edit path $P'$ with $|P'| < |P|$ and $\mathrm{apply}(\mathrm{G}_1, \mathrm{P'}) \cong G_2$, contradicting $P$'s property of being a shortest edit path from $G_1$ to $G_2$.

Denote by

$$
d_{M2} = (1 - \lambda) \cdot |P| = (1 - \lambda) \cdot d_{12}
$$

the interpolation target ($\mathrm{IC}_2$). Using a similar argument as above, we obtain

$$
d(G_M, G_2) = \mathrm{round}(d_{M2})
$$

Optimality now follows since GEDs are always integer-valued and the GEDs of $\mathrm{round}(d_{M1})$ and $\mathrm{round}(d_{M2})$ obtained by Alg. 1 are the integer values closest to their targets $d_{M1}$ and $d_{M2}$, respectively.

Using the facts that $0 \leq d_{M1} \leq d_{12}$ and $d_{M2} = d_{12} - d_{M1}$, we obtain

$$
\begin{aligned}
\mathrm{AIE}(G_M) &= |d_{\mathrm{GED}}(G_M, G_1) - \lambda \cdot d_{\mathrm{GED}}(G_1, G_2)| + |d_{\mathrm{GED}}(G_M, G_2) - (1 - \lambda) \cdot d_{\mathrm{GED}}(G_1, G_2)| \\
&= |\mathrm{round}(d_{M1}) - d_{M1}| + |\mathrm{round}(d_{M2}) - d_{M2}| \\
&= |\mathrm{round}(d_{M1}) - d_{M1}| + |\mathrm{round}(d_{12} - d_{M1}) - (d_{12} - d_{M1})| \\
&= |\mathrm{round}(d_{M1}) - d_{M1}| + |d_{12} - \mathrm{round}(-d_{M1}) - d_{12} + d_{M1}| \\
&= |\mathrm{round}(d_{M1}) - d_{M1}| + |\mathrm{round}(-d_{M1}) + d_{M1}| \\
&= 2 \cdot |\mathrm{round}(d_{M1}) - d_{M1}| \\
&\leq 1,
\end{aligned}
$$

where the last equality is obtained by considering the two cases (a) $d_{M1}$ is rounded down and (b) $d_{M1}$ is rounded up separately. This proves the claimed interpolation error. $\qquad\square$

# G  INTERPOLATION ANALYSIS

## G.1  ADDITIONAL RESULTS FOR MEAN INTERPOLATION ERROR

Tab. 11 contains results for multiple mixup ratios (in comparison to Tab. 4 from the main section which only includes a summary). We found that for GED-Mixup, the mIE was stable across different mixup ratios, whereas for other methods (such as If-Mixup, SubMix or S-Mixup), the mIE decreased with increasing mixup ratio.

Table 11: Detailed comparison of mean interpolation error (mIE) obtained by various methods and datasets (lower is better and $\geq 2$ is particularly bad). We were unable to generate graphs with FGW-Mixup on some datasets (denoted with –).

| Method | $\lambda$ | MUTAG | ENZYMES | IMDB-B | PROTEINS | IMDB-M | NCI1 | Avg. |
|---|---|---|---|---|---|---|---|---|
| GED-M. | Avg. | 0.05 | 0.01 | 0.01 | 0.01 | 0.03 | 0.03 | 0.02 |
|  | 0.5 | 0.06 | 0.01 | 0.02 | 0.02 | 0.03 | 0.03 | 0.03 |
|  | 0.8 | 0.04 | 0.01 | 0.01 | 0.01 | 0.03 | 0.03 | 0.02 |
|  | 0.9 | 0.04 | 0.01 | 0.01 | 0.01 | 0.02 | 0.03 | 0.02 |
| SubMix | Avg. | 0.45 | 0.28 | 0.49 | 0.30 | 0.53 | 0.37 | 0.40 |
|  | 0.5 | – | 0.53 | 0.94 | 0.57 | 0.99 | 0.64 | 0.73 |
|  | 0.8 | 0.62 | 0.21 | 0.35 | 0.22 | 0.39 | 0.29 | 0.34 |
|  | 0.9 | 0.28 | 0.11 | 0.18 | 0.10 | 0.20 | 0.17 | 0.17 |
| If-Mixup* | Avg. | 1.57 | 0.69 | 0.86 | 1.00 | 0.31 | 1.03 | 0.91 |
|  | 0.5 | 1.91 | 0.74 | 1.64 | 1.26 | 0.45 | 1.24 | 1.21 |
|  | 0.8 | 1.31 | 0.72 | 0.61 | 0.91 | 0.28 | 1.04 | 0.81 |
|  | 0.9 | 1.49 | 0.61 | 0.33 | 0.82 | 0.21 | 0.81 | 0.71 |
| S-Mixup* | Avg. | 3.56 | 0.95 | 1.32 | 0.72 | 0.96 | 2.10 | 1.60 |
|  | 0.5 | 4.03 | 1.17 | 1.52 | 1.16 | 1.20 | 2.89 | 2.00 |
|  | 0.8 | 3.37 | 0.97 | 1.10 | 0.56 | 0.92 | 1.68 | 1.43 |
|  | 0.9 | 3.28 | 0.69 | 1.34 | 0.44 | 0.77 | 1.72 | 1.37 |
| GeoMix* | Avg. | 4.31 | 1.32 | 0.66 | 1.60 | 0.69 | 2.60 | 1.86 |
|  | 0.5 | 4.85 | 1.54 | 0.81 | 1.67 | 0.83 | 2.50 | 2.03 |
|  | 0.8 | 4.25 | 1.24 | 0.55 | 1.80 | 0.57 | 2.93 | 1.89 |
|  | 0.9 | 3.85 | 1.19 | 0.61 | 1.34 | 0.65 | 2.36 | 1.67 |
| FGW-M. | Avg. | 2.76 | – | – | – | – | – | 2.76 |
|  | 0.5 | 1.95 | – | – | – | – | – | 1.95 |
|  | 0.8 | 2.51 | – | – | – | – | – | 2.51 |
|  | 0.9 | 3.82 | – | – | – | – | – | 3.82 |

## G.2 ADDITIONAL EXAMPLES OF INTERPOLATION ERRORS

This section contains plots that visualize the interpolation error $\mathrm{mIE}$ for further datasets and mixup ratios $\lambda$. In line with the discussion of Fig. 3 in the main paper, GED-Mixup interpolates well between inputs. The second best results are often obtained by SubMix, If-Mixup, or S-Mixup (except on MUTAG). Tab. 4 in the main section and Tab. 11 provide summaries beyond the example mixup graphs shown in the plots.

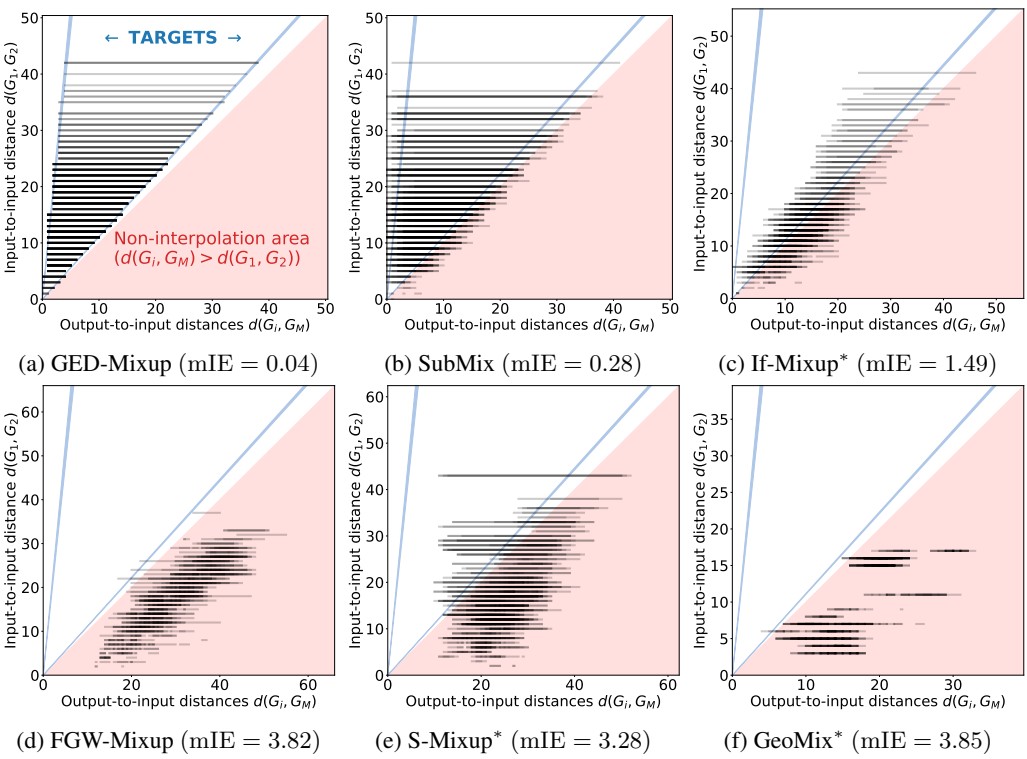

Figure 8: Visualization of mixup graphs produced on the MUTAG dataset ($\lambda = 0.9 \pm \varepsilon$).

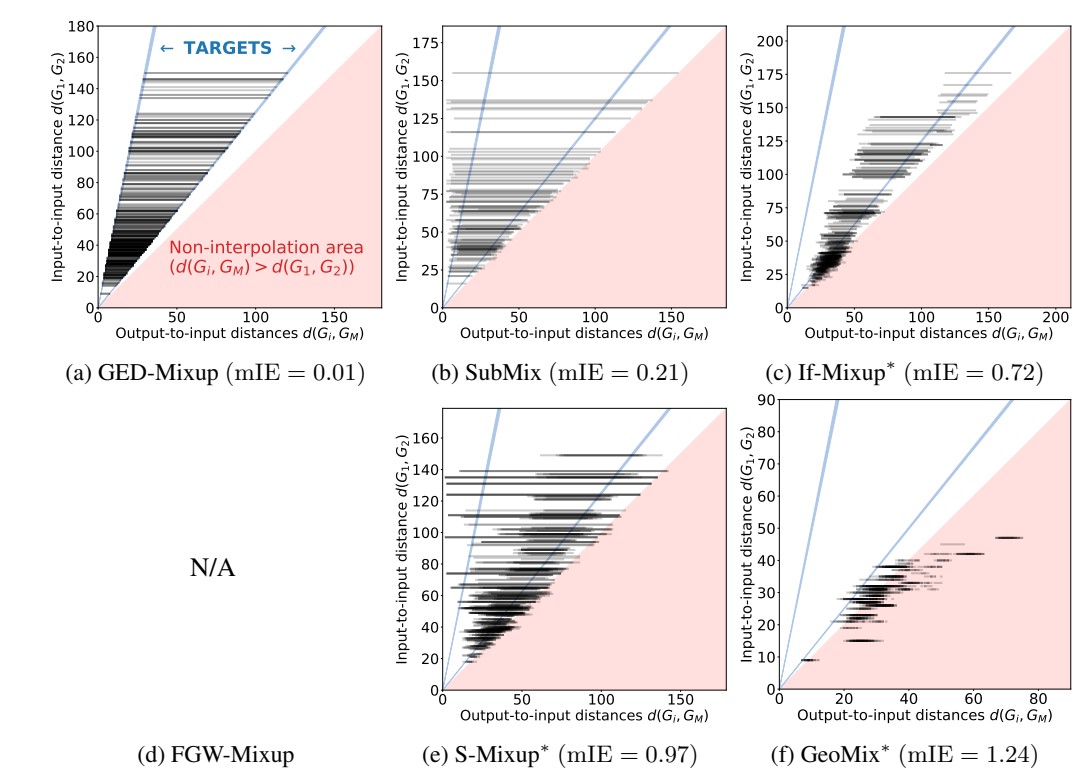

Figure 9: Visualization of mixup graphs produced on the ENZYMES dataset ($\lambda = 0.8 \pm \varepsilon$). Results for FGW-Mixup are missing since we were not able to generate graphs for ENZYMES.

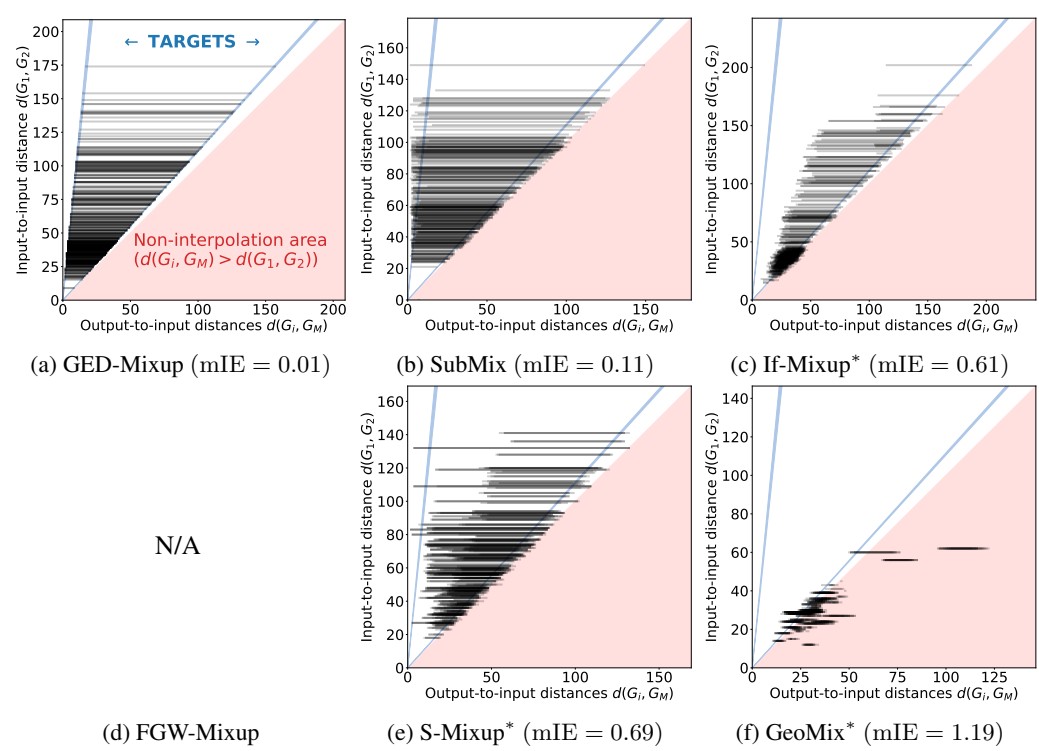

Figure 10: Visualization of mixup graphs produced on the ENZYMES dataset ($\lambda = 0.9 \pm \varepsilon$). Results for FGW-Mixup are missing since we were not able to generate graphs for ENZYMES.

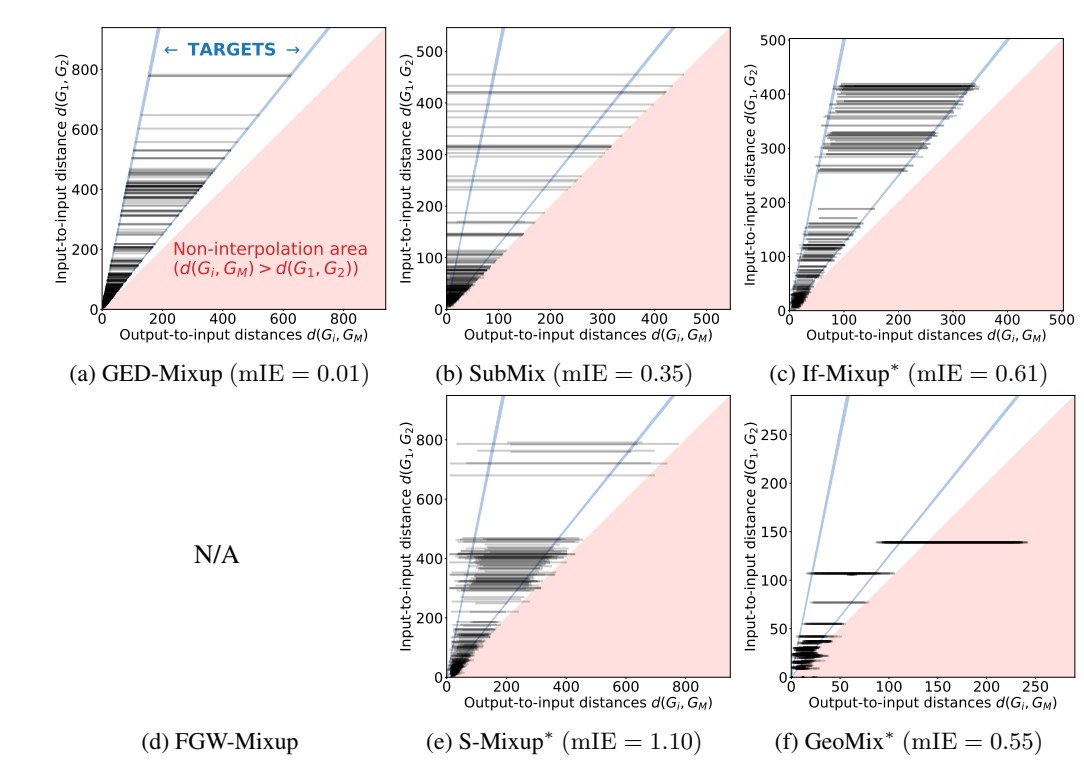

(a) GED-Mixup (mIE = 0.01)  (b) SubMix (mIE = 0.35)  (c) If-Mixup* (mIE = 0.61)

(d) FGW-Mixup  (e) S-Mixup* (mIE = 1.10)  (f) GeoMix* (mIE = 0.55)

Figure 11: Visualization of mixup graphs produced on the IMDB-BINARY dataset ($\lambda = 0.8 \pm \varepsilon$). Results for FGW-Mixup are missing since we were not able to generate graphs for IMDB-BINARY.

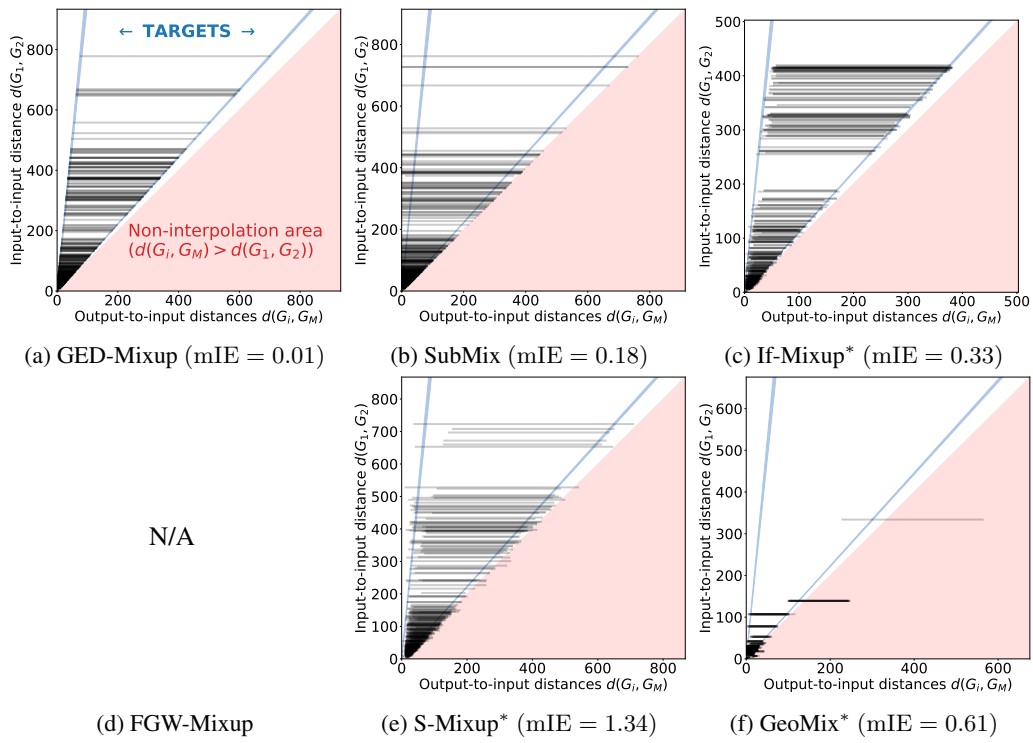

(a) GED-Mixup (mIE = 0.01)  (b) SubMix (mIE = 0.18)  (c) If-Mixup* (mIE = 0.33)

(d) FGW-Mixup  (e) S-Mixup* (mIE = 1.34)  (f) GeoMix* (mIE = 0.61)

Figure 12: Visualization of mixup graphs produced on the IMDB-BINARY dataset ($\lambda = 0.9 \pm \varepsilon$). Results for FGW-Mixup are missing since we were not able to generate graphs for IMDB-BINARY.

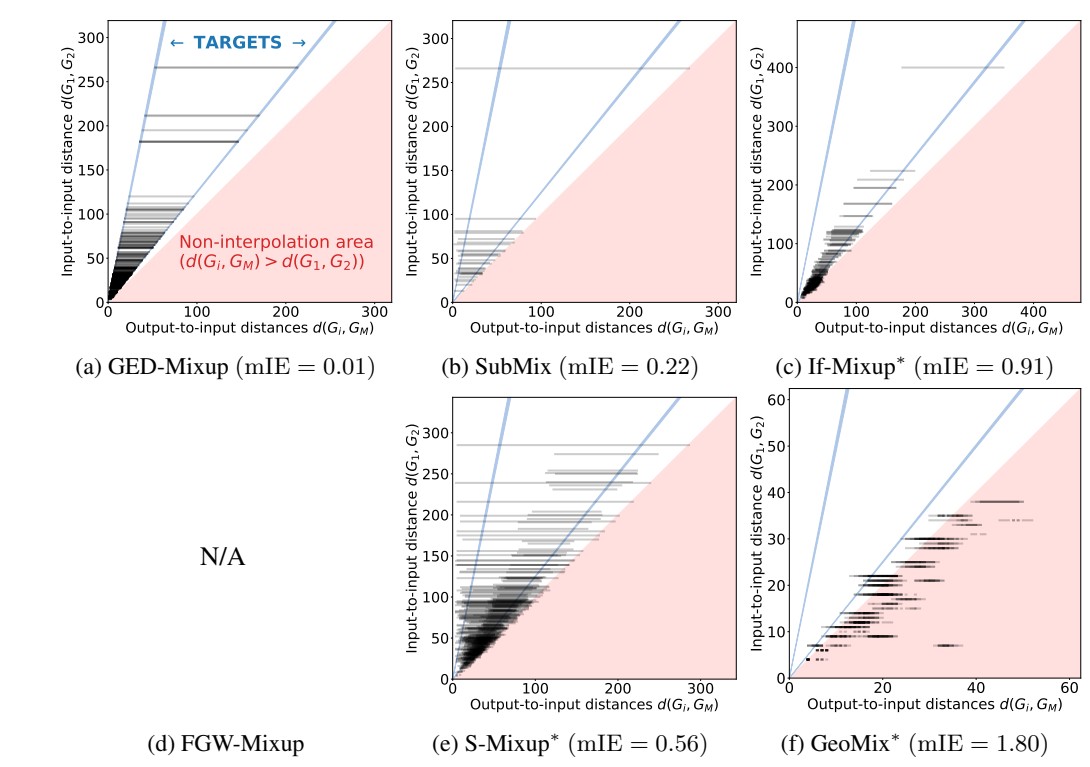

Figure 13: Visualization of mixup graphs produced on the PROTEINS dataset ($\lambda = 0.8 \pm \varepsilon$). Results for FGW-Mixup are missing since we were not able to generate graphs for PROTEINS.

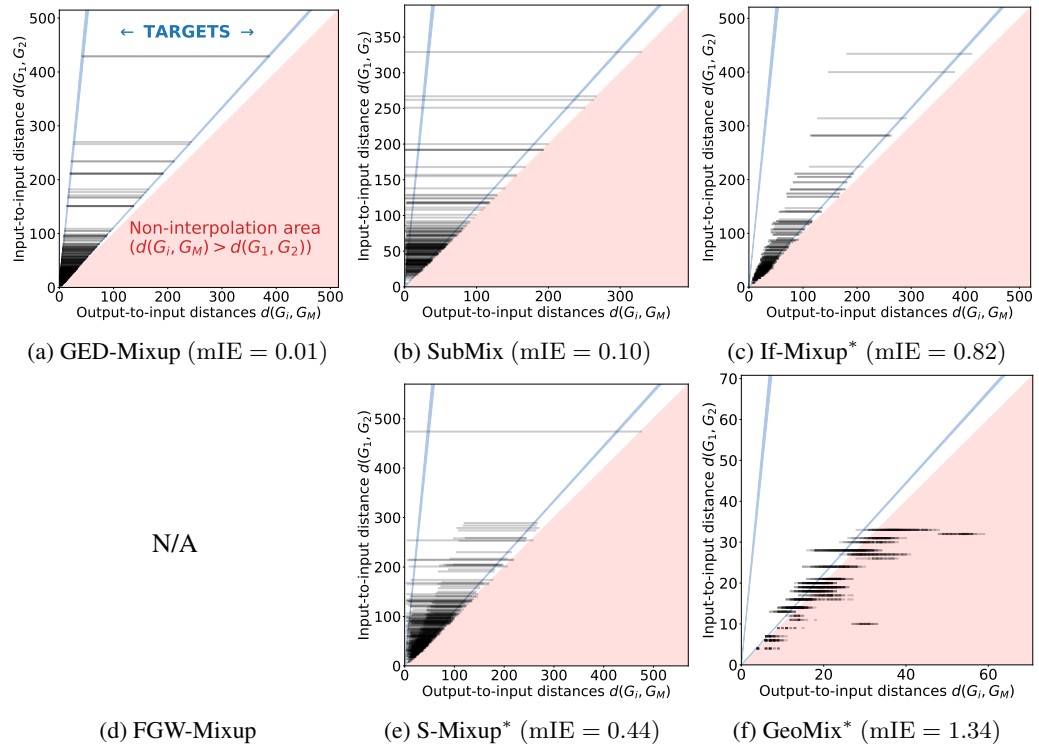

Figure 14: Visualization of mixup graphs produced on the PROTEINS dataset ($\lambda = 0.9 \pm \varepsilon$). Results for FGW-Mixup are missing since we were not able to generate graphs for PROTEINS.

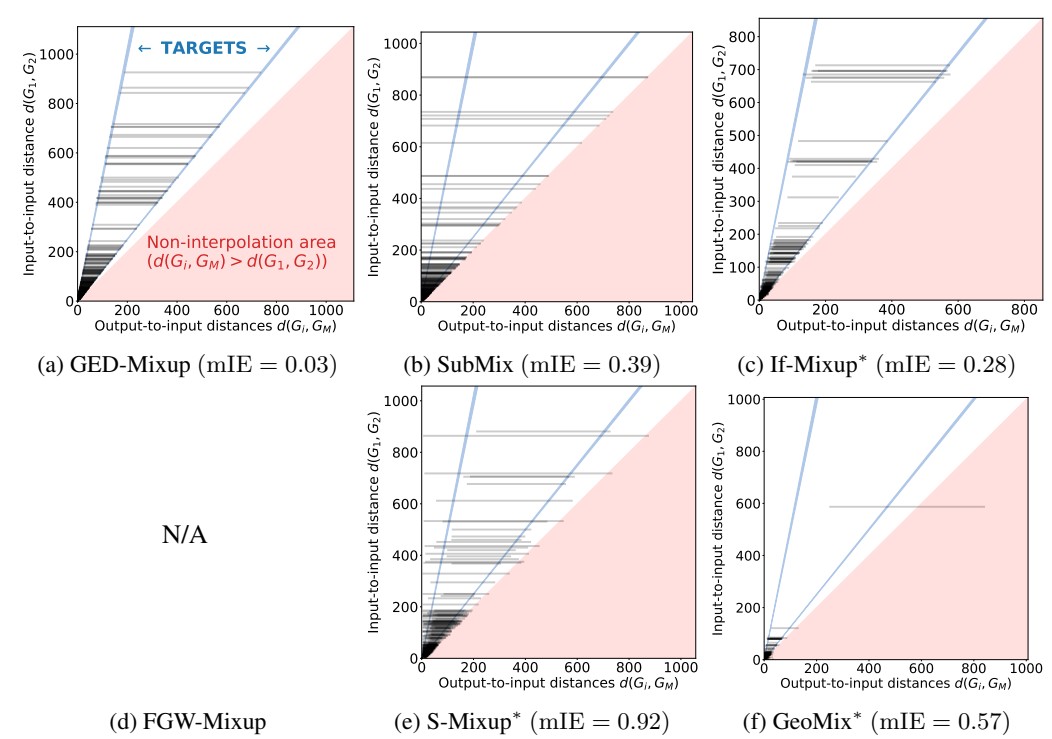

Figure 15: Visualization of mixup graphs produced on the IMDB-MULTI dataset ($\lambda = 0.8 \pm \varepsilon$). Results for FGW-Mixup are missing since we were not able to generate graphs for IMDB-MULTI.

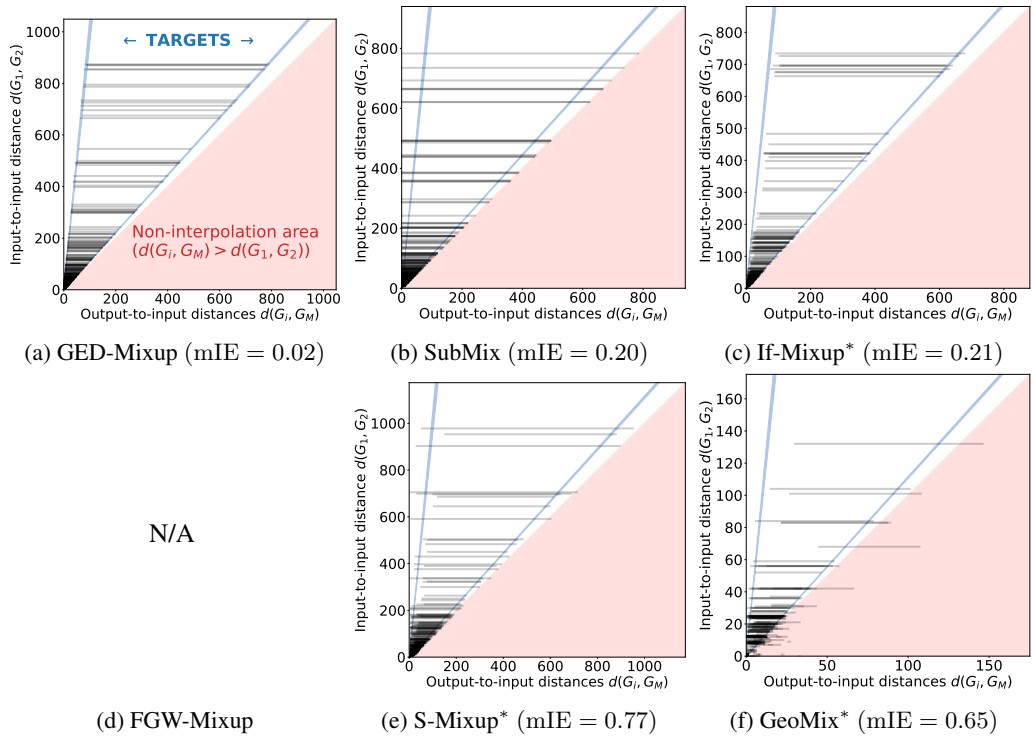

Figure 16: Visualization of mixup graphs produced on the IMDB-MULTI dataset ($\lambda = 0.9 \pm \varepsilon$). Results for FGW-Mixup are missing since we were not able to generate graphs for IMDB-MULTI.

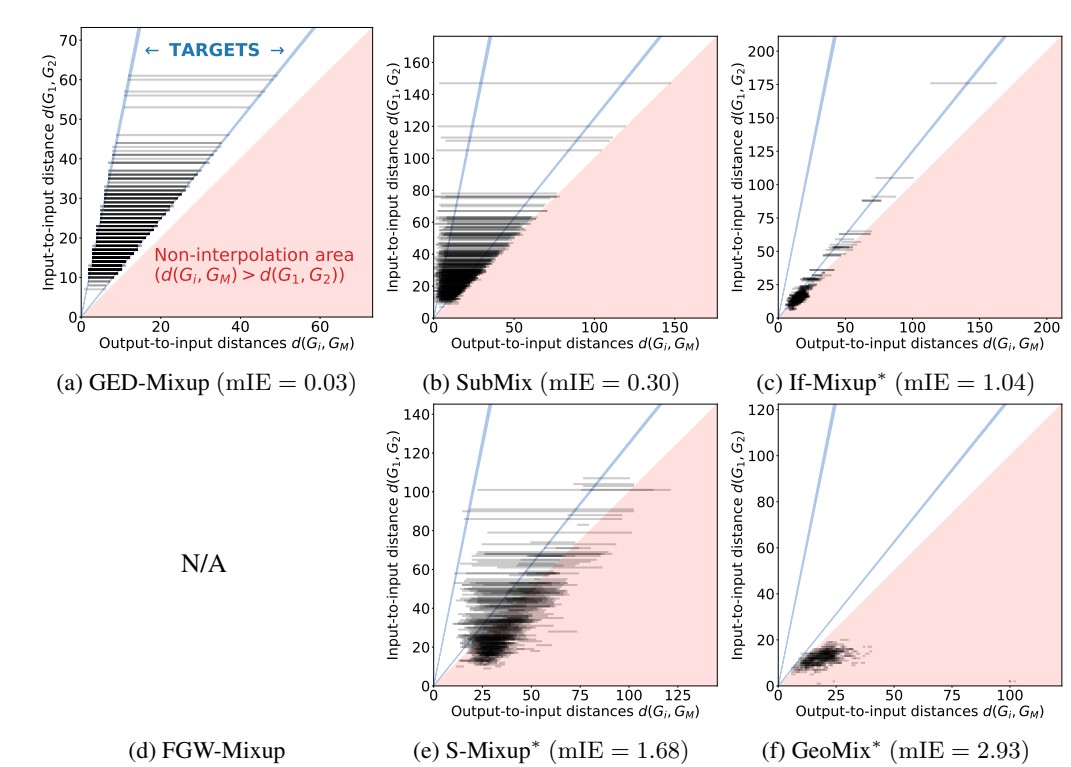

Figure 17: Visualization of mixup graphs produced on the NCI1 dataset ($\lambda = 0.8 \pm \varepsilon$). Results for FGW-Mixup are missing since we were not able to generate graphs for NCI1.

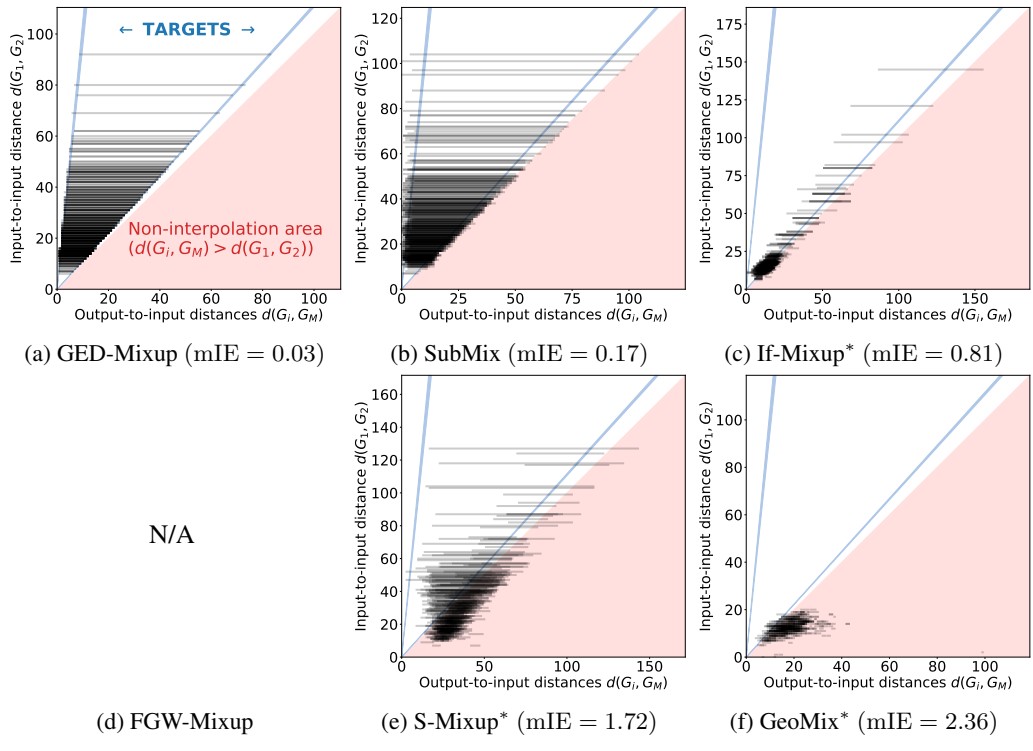

Figure 18: Visualization of mixup graphs produced on the NCI1 dataset ($\lambda = 0.9 \pm \varepsilon$). Results for FGW-Mixup are missing since we were not able to generate graphs for NCI1.

