# OpenReview forum: "Is Graph Mixup Beneficial? Investigating Interpolation And Empirical Performance of Graph Mixup Methods"
_ICLR.cc/2026/Conference — Submitted to ICLR 2026_

### Official Review · Reviewer_z4K1 · 2025-10-28

**Soundness:** 2
**Presentation:** 2
**Contribution:** 2
**Rating:** 4
**Confidence:** 5

**Summary:**

The work focuses on several interesting questions regarding graph mixup problems and methods: Does the interpolation align with the graph edit distance (GED)? Does misalignment lead to poor empirical performance? Does GED-aligned interpolation improve performance compared to the baseline? The paper first reviews different graph mixup methods, ranging from structure-based to embedding-based approaches, and then conducts empirical analyses to examine the benefits of mixup. Next, it analyzes the cause of performance differences by comparing interpolation errors with GED. The results are interesting. In summary, this is an engaging direction, and the paper provides valuable insights related to GED. However, it deserves clearer presentation to highlight its contributions—such as whether GED has been used before to evaluate or analyze mixup, whether GED-based mixup is newly proposed in this work, and what the generalizability and limitations of GED and the paper’s conclusions are in graph tasks.

**Strengths:**

1. The work focuses on several interesting questions regarding graph mixup problems and methods and is organized to address them step by step, making the paper relatively easy to follow.

2. The idea of GED-aligned mixup is interesting.

**Weaknesses:**

## 1. Evaluation tasks should be extended
One important application of mixup is data augmentation to address the issue of limited training data. This is a common scenario for molecular tasks, which can naturally be modeled as graphs. However, the current work only includes four datasets, with only one molecular dataset, and it is unclear what the data distributions look like—whether the labeled data are imbalanced and whether data augmentation is truly needed.

Conclusions such as “even optimal interpolation did not lead to performance improvement” and “graph mixup provided no significant improvement over the no-mixup baseline” are too arbitrary and dataset-specific. It is important to ensure that these conclusions are generalizable across different types of graph learning problems.

To make the conclusions more robust and convincing, the study should be extended to broader molecular benchmarks, such as those from OGB, MoleculeNet, or Polaris. Many molecular tasks are regression problems, which is another missing aspect in the paper’s current analysis. The tasks should also explicitly consider issues related to data imbalance and small-data regimes.

## 2. Insufficient coverage of existing mixup algorithms

The set of reviewed and evaluated mixup algorithms appears limited. Given the strong statement that “graph mixup provided no significant improvement over the no-mixup baseline, which questions the practical benefits of graph mixup,” several aspects remain unclear: What exactly is the no-mixup baseline? Which mixup variants are currently evaluated, and do they represent all major categories of mixup methods? Currently, only one embedding-based mixup approach is included. For a more systematic and fair evaluation, additional baselines should be considered. For example, there exists a line of research on graph rationale-based methods, which share conceptual similarities with mixup (e.g., selectively preserving interpretable parts of graphs). Including such methods would make the empirical comparison more comprehensive.


## 3. Limitations and generalizability of the mixup algorithms

Different mixup methods may vary in efficiency. How efficient is the GED-based mixup? An analysis of the training time and a comparison with other baselines should be included.

Another question concerns how different mixup methods generalize to molecular and other graph-structured data, since different graphs may have different node and edge features. When performing structural mixup, do the authors also mix the feature dimensions of nodes and edges? How are these aspects handled, and to what extent do these design choices affect the model’s performance?

**Questions:**

1. Is the GED-Mixup method newly proposed in this paper? How does it differ from or simplify the EPIC method?

2. How much computation time does the GED-Mixup process require? Is the runtime related to the dataset size or the graph size? How long does each training epoch take with this method?

3. Can GED-Mixup be applied to molecular graphs? Specifically, can it mix nodes and edges that contain multiple features, including both discrete and continuous values?

4. What types of atom and bond features are used in the MUTAG dataset?

---

> ### Author Response · Authors · 2025-11-14
> **Response to Review of Reviewer z4K1**
>
> Thank you for your insightful, constructive comments and your support of our work!
>
> ## Weaknesses
>
> ### W1
>
> > One important application of mixup is data augmentation to address the issue
> > of limited training data. This is a common scenario for molecular tasks, which
> > can naturally be modeled as graphs. However, the current work only includes
> > four datasets, with only one molecular dataset, and it is unclear what the
> > data distributions look like—whether the labeled data are imbalanced and
> > whether data augmentation is truly needed.
>
> Please see FAQ-3. We will also expand on the dataset description in the
> revision.
>
> > Conclusions such as “even optimal interpolation did not lead to performance
> > improvement” and “graph mixup provided no significant improvement over the
> > no-mixup baseline” are too arbitrary and dataset-specific. It is important to
> > ensure that these conclusions are generalizable across different types of
> > graph learning problems.
>
> Both of these statements are w.r.t. to the experiments and statistical analyses
> that we performed. Please see FAQ-1, FAQ-2 and FAQ-3.
>
> > To make the conclusions more robust and convincing, the study should be
> > extended to broader molecular benchmarks, such as those from OGB, MoleculeNet,
> > or Polaris. Many molecular tasks are regression problems, which is another
> > missing aspect in the paper’s current analysis. The tasks should also
> > explicitly consider issues related to data imbalance and small-data regimes.
>
> Please see FAQ-3.
>
> Thank you for the suggestion about including label distributions; we'll add and
> discuss the corresponding statistics in our revision.
>
> ### Weakness W2
>
> > The set of reviewed and evaluated mixup algorithms appears limited. Given the
> > strong statement that “graph mixup provided no significant improvement over
> > the no-mixup baseline, which questions the practical benefits of graph mixup,”
> > several aspects remain unclear: What exactly is the no-mixup baseline?
>
> The "no-mixup baseline" refers to training the graph neural network without
> performing any mixup. This is the baseline that mixup methods aim to improve on.
>
> To obtain this baseline (but also the results for all of the mixup methods), we
> followed unified, solid, and statistically sound experimental setup (including
> extensive hyperparameter optimization, nested cross validation, and statistical
> analysis). More details are described in App. B.
>
> > Which mixup variants are currently evaluated, and do they represent all major
> > categories of mixup methods? Currently, only one embedding-based mixup
> > approach is included.
>
> The variants are summarized in Table 1. To the best of our knowledge, this
> includes all major categories of graph mixup methods that were proposed in the
> literature.
>
> Embedding mixup is the goto-approach for embedding-based mixup, also in other
> domains. We'd appreciate pointers to other relevant embedding-based graph mixup
> methods that we may have missed.
>
> > For a more systematic and fair evaluation, additional baselines should be
> > considered. For example, there exists a line of research on graph
> > rationale-based methods, which share conceptual similarities with mixup (e.g.,
> > selectively preserving interpretable parts of graphs). Including such methods
> > would make the empirical comparison more comprehensive.
>
> Thanks! We will look into these methods during the revision period and report
> back. Generally, we, our study, and consequently, our findings focus on graph
> mixup.
>
> ### Weakness W-3
>
> > Different mixup methods may vary in efficiency. How efficient is the GED-based
> > mixup? An analysis of the training time and a comparison with other baselines
> > should be included.
>
> Please see FAQ-4.
>
> > Another question concerns how different mixup methods generalize to molecular
> > and other graph-structured data, since different graphs may have different
> > node and edge features.
>
> Please see FAQ-2.
>
> > When performing structural mixup, do the authors also mix the feature
> > dimensions of nodes and edges? How are these aspects handled, and to what
> > extent do these design choices affect the model’s performance?
>
> Generally yes.
>
> More specifically, the basic options are (i) mixup node/edge features as well or
> (ii) use the node/edge features of one of the input graphs. For all prior
> methods, we used the original implementations, each of which is tied to a
> specific form of feature mixup. For GED-Mixup, we simply used (ii) via the
> label-substitution edit operation (see Alg. 1).
>
> That being said, it may indeed be beneficial to explore feature mixup further,
> and it's a suitable direction for further research. The focus of our current
> work, however, is in exploring existing methods rather than proposing new ones.
>
> ---
>
> Please find our answers to your questions in the other comment.

---

> > ### Author Response · Authors · 2025-11-14
> > **Response to Review of Reviewer z4K1 (Part II)**
> >
> > ## Questions
> >
> > ### Q1
> >
> > > Is the GED-Mixup method newly proposed in this paper? How does it differ from
> > > or simplify the EPIC method?
> >
> > GED-Mixup hasn't been proposed before, but (i) we only use it as an analysis
> > tool and (ii) the method is very natural. We do not claim any credit for this
> > method.
> >
> > EPIC is significantly more involved than GED-Mixup. It uses learned edit cost
> > models, and it approximates graph edit distances. Both of these properties are
> > not suited to our analysis goals. We did not consider EPIC in our experimental
> > study as source code was not available to us.
> >
> > ### Q2
> >
> > > How much computation time does the GED-Mixup process require? Is the runtime
> > > related to the dataset size or the graph size? How long does each training
> > > epoch take with this method?
> >
> > Thank you! We will provide these results in the revision. Please see FAQ-4.
> >
> > ### Q3
> >
> > > Can GED-Mixup be applied to molecular graphs? Specifically, can it mix nodes
> > > and edges that contain multiple features, including both discrete and
> > > continuous values?
> >
> > Yes. But again, our goal is not to suggest that GED-Mixup is a suitable mixup
> > method for applications, but to use it as an analysis tool.
> >
> > ### Q4
> >
> > > What types of atom and bond features are used in the MUTAG dataset?
> >
> > Thank you. We use all of the seven available node features in TUDataset (we set
> > `use_node_attr` to `True` for all datasets, including MUTAG), but failed to
> > point this out correctly in Tab. 5 (which we will revise).

---

> ### Author Response · Authors · 2025-11-25
>
> Thank you for the suggestions made in your review. We have conducted
> additional experiments following these suggestions; the results are
> reported in a separate post. We will include these (and other) changes
> in a revised version of the paper by Dec 2. We greatly appreciate if you
> let us know whether or not these changes address your concerns and, if
> not, which additional changes you consider important. Your feedback is
> helpful and important to us.

---

> > ### Comment · Reviewer_z4K1 · 2025-11-27
> >
> > Thank you for the response. Many of my initial points are acknowledged, but several concerns remain unresolved. My main issue is that the current evaluation of mixup methods lacks a clear notion of “context,” which leads to claims about usefulness or lack of usefulness that are not well supported. For example, in a dataset with both noisy and clean samples, using the noisy subset for mixup can naturally degrade performance, while using the clean subset may produce the opposite outcome. Without a clear definition of these contexts, it is difficult to interpret the results.
> >
> > Related to this, the paper presents several datasets, but the analysis does not clarify whether they reflect practical settings. Key aspects such as the level of noise, imbalance, or real-world relevance are not sufficiently examined. Although additional results were included, it remains unclear to what degree they are realistic or informative.
> >
> > For example, MUTAG contains only 188 nitroaromatic compounds. It is not representative of current molecular learning tasks. Even widely used benchmarks such as OGB have received extensive criticism. I do not require the use of specific datasets, but more realistic datasets with explicit context would support stronger conclusions.
> >
> > Finally, the authors mention several planned updates, but these changes are not yet reflected in the manuscript. The paper would benefit from revisions that clarify dataset choice, methodology, and the scope of the conclusions to fully resolve the original concerns.

---

> > > ### Author Response · Authors · 2025-11-28
> > >
> > > Thank you for your quick response!
> > >
> > > > My main issue is that the current evaluation of mixup methods lacks a clear
> > > > notion of “context,” which leads to claims about usefulness or lack of
> > > > usefulness that are not well supported. For example, in a dataset with both
> > > > noisy and clean samples, using the noisy subset for mixup can naturally
> > > > degrade performance, while using the clean subset may produce the opposite
> > > > outcome. Without a clear definition of these contexts, it is difficult to
> > > > interpret the results.
> > >
> > > In a nutshell, our study suggest that there is currently little positive
> > > evidence in favor of mixup: the benefits of graph mixup as suggested in prior
> > > work did not materialize in our unified, comprehensive, and sound experimental
> > > setup. We do not intend to claim (and will carefully revisit the paper to make
> > > sure that we do not actually claim) that mixup cannot be beneficial.
> > >
> > > That being said, we will add a discussion to the paper more clearly arguing on
> > > the relationship between mixup, interpolation, and performance using informative
> > > examples. We will make an argument that mixup can produce very good results in
> > > principle, and likewise, that non-interpolating (and thus "not-really-mixup")
> > > methods may also improve quality. Finally, we will more clearly discuss the
> > > goals of our study along the lines of the FAQ.
> > >
> > > As for context, would you mind clarifying your concerns and which insights are
> > > missing? Note that we use cross validation for training, i.e., every data point
> > > is treated "equally" and used to train/validate/test in some folds. For example,
> > > any label noise would affect all methods equally.
> > >
> > > > Related to this, the paper presents several datasets, but the analysis does
> > > > not clarify whether they reflect practical settings. Key aspects such as the
> > > > level of noise, imbalance, or real-world relevance are not sufficiently
> > > > examined. Although additional results were included, it remains unclear to
> > > > what degree they are realistic or informative.
> > >
> > > We acknowledge that we did not study some of these aspects (and prior work on
> > > mixup did not study these either, to the best of our knowledge). As we noted
> > > above, our study found little positive evidence in favor mixup. We do not intend
> > > to claim, however, that mixup may not be beneficial in general.
> > >
> > > As for level of noise, we have conducted an additional experiment, as you noted,
> > > suggesting that---within in the limitations of that experiment---mixup did not
> > > help to combat label noise. We acknowledge that this experiment on its own is
> > > clearly too limited, but it does not provide any promising results in favor of
> > > mixup either.
> > >
> > > As for class imbalance, we acknowledge that we did not explore this question. We
> > > will provide more information about the datasets that we used, but we otherwise
> > > would like to leave this question for future work. The scope of our experiments
> > > is already substantial, and it's not always feasible to answer all possible
> > > questions.
> > >
> > > Finally, real-world relevance is a concern that may be hard to address in the
> > > context of a scientific study. Since none of the mixup methods provided
> > > significant benefits on the benchmark datasets that we used, we feel
> > > that---whether or not these datasets qualify as real-world relevant---, at the
> > > very least, graph mixup does not appear to be as promising as suggested in prior
> > > work.

---

> > > > ### Author Response · Authors · 2025-11-28
> > > >
> > > > > I do not require the use of specific datasets, but more realistic datasets
> > > > > with explicit context would support stronger conclusions.
> > > >
> > > > The reasons why we considered the particular datasets used in our study, and why
> > > > we can't really consider the OGB datasets, are outlined in FAQ-3 (and we will
> > > > clarify these points in the revision). The most relevant points are:
> > > >
> > > > 1. They are frequently used in the existing literature on graph mixup.
> > > >
> > > > 2. The experimental study is computationally feasible. In particular, methods
> > > >    such as FGW-Mixup and GeoMix have substantial computational cost, but we
> > > >    wanted to include these methods.
> > > >
> > > > 3. A key motivation of using graph mixup is to deal with data scarcity, i.e.,
> > > >    relatively small training set sizes for complex tasks.
> > > >
> > > > Consider, for example, the ogbg-code2 dataset (452k graphs, mean size 125), even
> > > > though you also considered this dataset questionable. Using such a dataset
> > > > violates:
> > > >
> > > > 1. The actual literature of graph mixup did not consider this dataset.
> > > >
> > > > 2. The experimental study would not be feasible. We considered and tuned many
> > > >    different methods, which has a substantial cost even for small datasets. For
> > > >    instance, our new IMDB-MULTI results for GCN already required more than 300
> > > >    GPU-hours.
> > > >
> > > > 3. As training set size is large, we'd leave the data scarcity terrain.
> > > >
> > > > What are your thoughts on these arguments? Do you have any suggestions for us on
> > > > how to proceed?
> > > >
> > > > > Finally, the authors mention several planned updates, but these changes are
> > > > > not yet reflected in the manuscript. The paper would benefit from revisions
> > > > > that clarify dataset choice, methodology, and the scope of the conclusions to
> > > > > fully resolve the original concerns.
> > > >
> > > > Absolutely. We focused on expanding the experimental study and discussions with
> > > > you first. All the points we made here will be part of the revised paper. When
> > > > uploading the revision, we will post clearly what we changed to address you
> > > > concerns and improve the paper. We feel that the feedback we obtained was highly
> > > > valuable for a revision, and hopefully leads to a significantly improved
> > > > revision.

---

### Official Review · Reviewer_CTcZ · 2025-10-30

**Soundness:** 3
**Presentation:** 3
**Contribution:** 3
**Rating:** 4
**Confidence:** 3

**Summary:**

This paper revisits graph mixup techniques for graph classification. It conducts an empirical evaluation of several state-of-the-art graph mixup methods and analyzes their behavior through the lens of graph edit distance. Through both statistical testing and interpolation-based analysis, the study finds that current graph mixup methods provide no significant improvement over the no-mixup baseline, even when interpolation quality is high.

**Strengths:**

- The paper provides a unified and principled analysis of various graph mixup methods using edit distance as a common framework, which is conceptually elegant and insightful.
- The paper is well written and easy to follow. The discussion of related work is clear and well organized, providing an informative overview of existing graph mixup approaches.

**Weaknesses:**

- The experimental scope appears limited. The study evaluates only four datasets and contain relatively small graphs. This dataset selection may make it difficult to generalize the findings to other graph domains. Since the subsequent analyses rely heavily on these empirical results, expanding the dataset diversity would greatly strengthen the study’s conclusions.
- Figure 2 provides intriguing evidence that lower mIE values are associated with better accuracy; however, this relationship remains somewhat inconclusive, as different mixup methods vary in several aspects beyond mIE. To better support the argument, the authors could run an ablation study using GED-Mixup. They could create edit sets that are not perfectly optimal (with higher mIE values) and gradually adjust how suboptimal they are. Observing how performance changes in this setting would help clarify whether mIE actually affects accuracy.
- The domain composition of the datasets also warrants consideration. GED is particularly appropriate for molecular or bioinformatics graphs, where small structural edits correspond to meaningful chemical or biological variations. This might partly explain why interpolation quality correlates strongly with performance in these datasets. In contrast, on the IMDB-BINARY social-network dataset, methods with low mIE (e.g., GED-Mixup, SubMix, If-Mixup) do not exhibit clear performance advantages, suggesting that the observed trend could be domain-specific rather than universal.

**Questions:**

- The study evaluates only four small-scale TUDataset benchmarks. How confident can we be that the findings generalize to larger or more diverse graph domains, such as molecular graphs with higher node counts, biochemical interaction networks, or large social networks?
- Results on the IMDB-BINARY dataset appear inconsistent with the findings from molecular datasets. Is the link between interpolation fidelity and downstream accuracy dependent on domain semantics, such as chemically meaningful edit operations?

---

> ### Author Response · Authors · 2025-11-14
> **Response to Review of Reviewer CTcZ**
>
> Thank you for your insightful, constructive comments and your support of our work!
>
> ## Weaknesses
>
> ### W1
>
> > The experimental scope appears limited. The study evaluates only four datasets
> > and contain relatively small graphs. This dataset selection may make it
> > difficult to generalize the findings to other graph domains. Since the
> > subsequent analyses rely heavily on these empirical results, expanding the
> > dataset diversity would greatly strengthen the study’s conclusions.
>
> Please see FAQ-3.
>
> ### Weakness W-2
>
> > Figure 2 provides intriguing evidence that lower mIE values are associated
> > with better accuracy; however, this relationship remains somewhat
> > inconclusive, as different mixup methods vary in several aspects beyond mIE.
>
> Please see FAQ-1.
>
> > To better support the argument, the authors could run an ablation study using
> > GED-Mixup. They could create edit sets that are not perfectly optimal (with
> > higher mIE values) and gradually adjust how suboptimal they are. Observing how
> > performance changes in this setting would help clarify whether mIE actually
> > affects accuracy.
>
> Please see FAQ-2.
>
> ### Weakness W-3
>
> > The domain composition of the datasets also warrants consideration. GED is
> > particularly appropriate for molecular or bioinformatics graphs, where small
> > structural edits correspond to meaningful chemical or biological variations.
> > This might partly explain why interpolation quality correlates strongly with
> > performance in these datasets.
>
> We did not find nor did we intend to claim strong correlation between
> interpolation quality and performance. Please see FAQ-1.
>
> > In contrast, on the IMDB-BINARY social-network dataset, methods with low mIE
> > (e.g., GED-Mixup, SubMix, If-Mixup) do not exhibit clear performance
> > advantages, suggesting that the observed trend could be domain-specific rather
> > than universal.
>
> Please see above and FAQ-1.
>
> That being said, in the revision, we plan to include an additional dataset in
> the social network domain to further explore this point.
>
> ## Questions
>
> ### Q1
>
> > The study evaluates only four small-scale TUDataset benchmarks. How confident
> > can we be that the findings generalize to larger or more diverse graph
> > domains, such as molecular graphs with higher node counts, biochemical
> > interaction networks, or large social networks?
>
> First, note that our datasets include graphs with >500 vertices and >1000 edges.
> These graph sizes cover a substantial fraction of the use cases for graph
> classification, but, as you argue, not all of them.
>
> We assess this question to the extent possible given the data via assumption
> (A1), i.e., the estimated performance of graph mixup on other datasets. The
> results in Table 3 indicate that none of the results under (A1) are
> statistically significant. We cannot be confident that graph mixup provides or
> does not provide benefits (or, even in case it is beneficial, that effect sizes
> are substantial).
>
> ### Q2
>
> > Results on the IMDB-BINARY dataset appear inconsistent with the findings from
> > molecular datasets. Is the link between interpolation fidelity and downstream
> > accuracy dependent on domain semantics, such as chemically meaningful edit
> > operations?
>
> Please see answer to W3 above.

---

> ### Author Response · Authors · 2025-11-25
>
> Thank you for the suggestions made in your review. We have conducted
> additional experiments following these suggestions; the results are
> reported in a separate post. We will include these (and other) changes
> in a revised version of the paper by Dec 2. We greatly appreciate if you
> let us know whether or not these changes address your concerns and, if
> not, which additional changes you consider important. Your feedback is
> helpful and important to us.

---

> > ### Comment · Reviewer_CTcZ · 2025-11-27
> >
> > Thank you for the additional clarification in the rebuttal. However, some of the central issues still appear insufficiently clarified.
> >
> > - First, although two additional datasets from TUDataset were included, these datasets also contain a small number of training graphs. It would be more convincing to include larger-scale benchmarks such as OGB. Regarding node counts, the authors note that some graphs contain more than 500 nodes, but the average number of nodes in the selected datasets remains around 20–40. This suggests that only a very small fraction of the graphs are actually large, which I do not consider sufficient. Since the main claims of the paper rely on empirical evidence, a broader and more diverse set of datasets is necessary.
> >
> > - Second, both the paper and the rebuttal (FAQ 1) argue that "bad interpolation properties are detrimental for the performance of graph mixup," but I believe this claim remains inadequately supported. While S-Mixup and GeoMixup exhibit higher mIE and lower performance, other factors besides mIE are not controlled, so their reduced performance cannot be attributed to interpolation quality alone. To properly establish a correlation between mIE and performance, it is necessary to control for confounding factors and vary only mIE while measuring performance. As suggested in my review, one possible way to check this is to perform mixup using suboptimal edit sets derived from GED. This would allow the authors to test whether intentionally degrading the interpolation quality leads to predictable changes in performance.

---

> > > ### Author Response · Authors · 2025-11-27
> > >
> > > Thank you for your prompt reply!
> > >
> > > We agree that the datasets used in our study are rather small and that their
> > > graphs are not too large. Moreover, as you argue, the OGB graph classification
> > > datasets have more training data and some of them also have larger graphs (up to
> > > a mean size of 243). We acknowledge that this is a limitation of our study.
> > >
> > > The reasons why we considered the particular datasets used in our study, and why
> > > we can't really consider the OGB datasets, are outlined in FAQ-3. The most
> > > relevant points are:
> > >
> > > 1. They are frequently used in the existing literature on graph mixup.
> > >
> > > 2. The experimental study is computationally feasible. In particular, methods
> > >    such as FGW-Mixup and GeoMix have substantial computational cost, but we
> > >    wanted to include these methods.
> > >
> > > 3. A key motivation of using graph mixup is to deal with data scarcity, i.e.,
> > >    relatively small training set sizes for complex tasks.
> > >
> > > Consider, for example, the ogbg-code2 dataset (452k graphs, mean size 125).
> > > Using such a dataset violates:
> > >
> > > 1. The actual literature of graph mixup did not consider this dataset.
> > >
> > > 2. The experimental study would not be feasible. We considered and tuned many
> > >    different methods, which has a substantial cost even for small datasets. For
> > >    instance, our new IMDB-MULTI results for GCN already required more than 300
> > >    GPU-hours.
> > >
> > > 3. As training set size is large, we'd leave the data scarcity terrain.
> > >
> > > What are your thoughts on these arguments? Do you have any suggestions for us on
> > > how to proceed?
> > >
> > > With respect to your second point, we consider doing such a controlled
> > > experiment, which may indeed be valuable, and report back ASAP.

---

> ### Author Response · Authors · 2025-11-28
>
> Thank you again for your suggestion to analyze the relationship between interpolation properties and empirical performance in a controlled setting.
>
> We performed such an experiment, and found that increasing mIE (and thus decreasing interpolation quality) led to a significant decline in accuracy, eventually resulting in a performance close to random guessing. This experiment supports our hypothesis that bad interpolation properties can be detrimental for empirical performance.
>
> **Results:**
>
> | Model | Dataset    | Method          | (Expected) mIE | Acc. ± SE (%)  |
> | :---- | :--------- | :-------------- | -------------: | :------------- |
> | GCN   | IMDB-MULTI | GED-M.          |           0.00 | 46.16 ± 0.76   |
> | GCN   | IMDB-MULTI | GED-M.          |           0.25 | 44.98 ± 1.02   |
> | GCN   | IMDB-MULTI | GED-M.          |           0.50 | 44.67 ± 0.83   |
> | GCN   | IMDB-MULTI | GED-M.          |           0.75 | 41.33 ± 1.12   |
> | GCN   | IMDB-MULTI | GED-M.          |           1.00 | 35.51 ± 1.92   |
> | N/A   | IMDB-MULTI | Random guessing |            N/A | $\approx$33.33 |
>
> **Experimental setup:**
>
> We considered one model (GCN) and one dataset (IMDB-MULTI), using the respective tuned hyperparameters. We focus on GED-Mixup in this experiment, as it allowed us to precisely control the interpolation error (i.e., mIE).
>
> We proceeded as in GED-Mixup, but used a different mixup ratio for input graphs and input labels. Effectively, this moves the mixup graph closer to a randomly chosen input graph than desired, and thus immediately violates the two interpolation targets in a controlled fashion.
>
> In more detail, given a desired interpolation error $\epsilon$ as input (expected mIE in the table above), instead of using interpolation ratio $\lambda$, we mix with $\lambda\pm\epsilon$ (clipped to [0,1]). Since GED-Mixup interpolates optimally (recall Prop. 2), this leads to an expected mIE of $\epsilon$.
>
> **Limitations:**
>
> One dataset and one model only. Moreover, all mixup graphs remain on an optimal edit path between the input graphs. This may be seen as a somewhat optimistic setting, yet it sufficed to observe degradation in empirical performance. Moreover, there are many ways to "deviate" from the interpolation targets; this is perhaps the simplest one. We used it here because it is both interpretable and feasible to implement as well as run.
>
> Is this what you had in mind and does it (at least partially) address your concerns?

---

### Official Review · Reviewer_uWeG · 2025-10-30

**Soundness:** 3
**Presentation:** 3
**Contribution:** 3
**Rating:** 6
**Confidence:** 4

**Summary:**

The paper analyzes whether graph mixup actually improves graph classification performance. It first conducts an empirical comparison of existing graph mixup methods against baselines on four benchmark datasets, finding that none achieve statistically significant improvements. To further test generality, the authors perform a pooled analysis across datasets and models, showing that even when aggregated, mixup methods fail to yield consistent benefits. They then introduce an interpolation-based analysis framework using the graph edit distance to quantify interpolation error and assess the quality of augmented graphs. Additionally, they propose a mixup method based on optimal graph alignment, termed GED-Mixup. Their results show that most existing methods have high interpolation error, and that while better interpolation tends to correlate with improved performance, even optimal interpolation does not lead to significant gains.

**Strengths:**

1. The paper presents an independent and systematic empirical evaluation of existing graph mixup methods using a unified experimental setup and pooled analysis. This contributes to a clearer understanding of the empirical effectiveness of mixup in graph classification, addressing inconsistencies in prior studies.

2. The introduction of an interpolation-based metric using graph edit distance is a useful addition, providing a quantitative way to assess how well mixup outputs interpolate between input graphs. This analysis helps connect structural properties of the generated graphs with their empirical performance - an aspect that has been largely overlooked in earlier work.

**Weaknesses:**

1. The paper shows that existing graph mixup methods do not yield significant performance gains; however, it remains unclear why mixup fails. The authors demonstrate that many methods produce poor interpolations, yet even optimal interpolation (via GED-Mixup) does not improve accuracy significantly (Figure 2). This raises an unanswered question about the underlying cause of mixup’s ineffectiveness. The interpretation would benefit from a deeper diagnostic analysis.

2. The paper also overemphasizes negative results without exploring other potential benefits of mixup. Prior work suggests that mixup can improve robustness to topology perturbations and label noise, but this study focuses solely on classification accuracy. A discussion or evaluation of such alternative objectives would provide a more balanced perspective and clarify whether mixup is universally ineffective or only for accuracy metrics.

3. The evaluation scope is limited to relatively small TU datasets. Including larger and more diverse benchmarks (e.g., Reddit, DD) would strengthen the conclusions and assess generalizability to real-world or large-scale graph settings.

4. The proposed GED-Mixup method is interesting but computationally impractical for larger graphs as mentioned in the paper. The paper does not discuss viable approximations or scalable alternatives, leaving open the question of how GED-based interpolation could be applied in realistic scenarios.

5. Finally, as a suggestion, it would be valuable to compare newer methods such as MomentMixup (which mixes graph moment vectors and may reduce interpolation error) and SIGL (which modifies alignment in G-Mixup). Evaluating these under the proposed interpolation framework could yield further insights into the design of effective graph mixup strategies.

**Questions:**

See weaknesses.

---

> ### Author Response · Authors · 2025-11-14
> **Response to Review of Reviewer uWeG**
>
> Thank you for your insightful, constructive comments and your support of our work!
>
> ## Weaknesses
>
> ### W1
>
> > The paper shows that existing graph mixup methods do not yield significant
> > performance gains; however, it remains unclear why mixup fails.
>
> In general, we agree with this statement. For some of the methods, however, our
> study suggests (with statistical significance) that mixup fails due to bad
> interpolation properties.
>
> > The authors demonstrate that many methods produce poor interpolations, yet
> > even optimal interpolation (via GED-Mixup) does not improve accuracy
> > significantly (Figure 2).
>
> Indeed. Please see FAQ-1.
>
> > This raises an unanswered question about the underlying cause of mixup’s
> > ineffectiveness. The interpretation would benefit from a deeper diagnostic
> > analysis.
>
> We agree. In the same spirit, there is also the unanswered question about why
> we'd expect graph mixup to be beneficial.
>
> One goal of this paper was to raise awareness for this issue. In contrast to the
> more intuitive arguments being made in some prior work, good interpolation did
> not imply improved downstream performance (and neither did it imply detrimental
> downstream performance).
>
> Further analysis is likely involved and method-specific, it's not immediately
> clear to us how to proceed, and it goes beyond the scope of our present work. We
> welcome suggestions about directions for deeper analysis though.
>
> ### W2
>
> > The paper also overemphasizes negative results without exploring other
> > potential benefits of mixup. Prior work suggests that mixup can improve
> > robustness to topology perturbations and label noise, but this study focuses
> > solely on classification accuracy. A discussion or evaluation of such
> > alternative objectives would provide a more balanced perspective and clarify
> > whether mixup is universally ineffective or only for accuracy metrics.
>
> Please see FAQ-2.
>
> ### W3
>
> > The evaluation scope is limited to relatively small TU datasets. Including
> > larger and more diverse benchmarks (e.g., Reddit, DD) would strengthen the
> > conclusions and assess generalizability to real-world or large-scale graph
> > settings.
>
> **Response:**
>
> Please see FAQ-3.
>
> ### W4
>
> > The proposed GED-Mixup method is interesting but computationally impractical
> > for larger graphs as mentioned in the paper. The paper does not discuss viable
> > approximations or scalable alternatives, leaving open the question of how
> > GED-based interpolation could be applied in realistic scenarios.
>
> Please see FAQ-4.
>
> ### W5
>
> > Finally, as a suggestion, it would be valuable to compare newer methods such
> > as MomentMixup (which mixes graph moment vectors and may reduce interpolation
> > error) and SIGL (which modifies alignment in G-Mixup). Evaluating these under
> > the proposed interpolation framework could yield further insights into the
> > design of effective graph mixup strategies.
>
> Thank you! We plan to discuss these works in the revision of our paper.

---

> ### Author Response · Authors · 2025-11-25
>
> Thank you for the suggestions made in your review. We have conducted
> additional experiments following these suggestions; the results are
> reported in a separate post. We will include these (and other) changes
> in a revised version of the paper by Dec 2. We greatly appreciate if you
> let us know whether or not these changes address your concerns and, if
> not, which additional changes you consider important. Your feedback is
> helpful and important to us.

---

### Official Review · Reviewer_BfwX · 2025-10-31

**Soundness:** 4
**Presentation:** 3
**Contribution:** 4
**Rating:** 8
**Confidence:** 4

**Summary:**

This paper proposes a graph mix-up method, a graph generator methodology which merges two input graphs, based on graph edit distance which is suitable for typically small graphs.

The authors use a novel evaluation methodology focused on interpolation error. The authors demonstrate that prior mixup works failed to generate graphs which interpolate between their inputs.

**Strengths:**

1. The authors demonstrate an important empirical finding in this research area, using a novel evaluation methodology. Prior mix-up works are fairly niche as a graph generative model, however, within this prior work the negative finding of structural coherence is very significant.

Furthermore, the presentation of the work is simple and understandable to a general AI research audience. This paper could be convincing for further work in this area.

2. The GED-mixup method is well motivated and suitable in the many domains for small graphs. This seems like a reasonable assumption, where higher controllability is be better suited for smaller graphs; large graph generation could be bracketed as work of a graph foundational model, this is fine.

3. The authors scope their research questions well and empirically support each of them. The two contributions (Sec 1) are significant.

**Weaknesses:**

1. Over-reliance on fidelity: the authors argue but don't demonstrate the utility of measures such as mIE. That is, what is the qualitative impact of methods with similar ACC but higher mIE, e.g. in Fig 2? Similarly, the authors don't present an evaluation of downstream robustness, e.g. for distribution shift, etc, which are the common use-cases for graph augmentation. The same critique is true: the graph generator distribution need not necessarily have good mIE if it adds robustness along another problem dimension.

2. The three levels of pooled analysis are difficult to follow and are not well reflected in the figures. e.g. is fig 2 representative under A2 assumptions? Is Fig 3 presented under A3? More space could be dedicated to contrasting results at these pooling levels.

**Questions:**

1. What is the downstream effect of mixup with high ACC and high mIE (e.g. Fig 2)?

2. Are there applications where high interpolation fidelity might not be best?

---

> ### Author Response · Authors · 2025-11-14
> **Response to Review of Reviewer BfwX**
>
> Thank you for your insightful, constructive comments and your support of our work!
>
> ## Weak points
>
> ### W1
>
> > Over-reliance on fidelity: the authors argue but don't demonstrate the utility
> > of measures such as mIE. That is, what is the qualitative impact of methods
> > with similar ACC but higher mIE, e.g. in Fig 2?
>
> Please see FAQ-1.
>
> > Similarly, the authors don't present an evaluation of downstream robustness,
> > e.g. for distribution shift, etc, which are the common use-cases for graph
> > augmentation. The same critique is true: the graph generator distribution need
> > not necessarily have good mIE if it adds robustness along another problem
> > dimension.
>
> Please see FAQ-2.
>
> ### W2
>
> > The three levels of pooled analysis are difficult to follow
>
> We will try to motivate and explain these methods better in the revised version,
> and we will address the points you raised below.
>
> > and are not well reflected in the figures. E.g. is Fig. 2 representative under
> > A2 assumptions?
>
> It is. Fig. 2 shows error bars under assumption A3 (the "shortest error bars").
> Under A1 and A2, the accuracy estimates will be the same, but the standard
> errors increase. We update the paper and add the corresponding figures under A1
> and under A2 as well.
>
> > Is Fig 3 presented under A3? More space could be dedicated to contrasting
> > results at these pooling levels.
>
> Fig. 3 does not show accuracy estimates, but analyzes the mixup graphs used
> during training. This analysis is independent from model evaluation and the
> assumptions used to compute standard errors. So in short: the figure is
> representative for A1, A2 and A3.
>
> ## Questions
>
> ### Q1
>
> > What is the downstream effect of mixup with high ACC and high mIE (e.g. Fig
> > 2)?
>
> See the response to W1.
>
> ### Q2
>
> > Are there applications where high interpolation fidelity might not be best?
>
> Again, see the response to W1. We conjecture: yes, there are such applications,
> but the interpolation fidelity likely needs to remain decent.

---

> ### Author Response · Authors · 2025-11-25
>
> Thank you for the suggestions made in your review. We have conducted
> additional experiments following these suggestions; the results are
> reported in a separate post. We will include these (and other) changes
> in a revised version of the paper by Dec 2. We greatly appreciate if you
> let us know whether or not these changes address your concerns and, if
> not, which additional changes you consider important. Your feedback is
> helpful and important to us.

---

### Author Response · Authors · 2025-11-14
**Frequently Asked Questions (FAQs): FAQ-1 & FAQ-2**

We'd like to thank all reviewers for their insightful, constructive comments and
their support of our work.

We'll address some of the most prominent thoughts being raised below, and we
plan to include these discussions into the revised version. Once we performed
the revision, we'll summarize all of our changes here.

We greatly appreciate feedback on these thoughts and suggestions.

### FAQ-1: Mixup methods with bad interpolation properties fell behind. What can we learn from good interpolation properties though?

Probably not much.

Our study suggests (with statistical significance) that bad interpolation
properties are detrimental for the performance of graph mixup, i.e., decent
interpolation appeared necessary. But, as you observed and our study also found,
good interpolation properties are clearly not sufficient and, consequently, the
mIE does not appear to be well-suited to compare methods with decent
interpolation properties (and we do not claim it to be).

In more detail, there are many ways to interpolate graphs such that the
interpolation error is decent. Roughly speaking, the entire green region in Fig.
1 of our paper can be seen as decent interpolation, but it's not clear which of
these interpolations is best w.r.t. downstream performance.

For example, GED-Mixup makes use of an optimal edit set. For each edit set,
however, there are many possible edit paths, some of which may be more suitable
than others for mixup. GED-Mixup simply chooses one of these paths at random
(subject to validity constraints), but this may indeed not be the most suitable
choice. We feel that an important direction for future work is to explore this
line of thought further; this is one of the key takeaways of our work.

### FAQ-2: The paper focuses on classification accuracy. What about other tasks or settings?

We followed the literature here.

The objective of most of the work on graph mixup is to improve graph
classification accuracy, and the primary experiments of these studies are
evaluating classification accuracy. The goal of our paper was to study these
methods in a unified, solid, and statistically sound experimental setup for the
task they were designed for.

The impact of graph mixup for other tasks and on other settings---such as
regression, robustness properties, or the ability to handle label noise---may be
different, indeed. While interesting and worth exploring, these directions
generally exceed the scope of this paper.

That being said, we will use the rebuttal period to try to shed some insight
into these points. A natural candidate for exploration is the impact of label
noise; we thus plan to focus on this aspect.

---

### Author Response · Authors · 2025-11-14
**Frequently Asked Questions (FAQs): FAQ-3 & FAQ-4**

### FAQ-3: The study uses four real-world datasets. What about other datasets?

We'd first like to highlight the rationale for choosing these four datasets:

1. They are frequently used in the existing literature on graph mixup.
2. They cover a variety of domains: social networks (IMDB), bioinformatics
   (ENZYMES, PROTEINS), molecular data (MUTAG).
3. The experimental study is computationally feasible. In particular, methods
   such as FGW-Mixup and GeoMix have substantial computational cost, but we
   wanted to include these methods.
4. Our goal was to use a unified, solid, and statistically sound experimental
   setup. This further enhances computational costs, as we performed extensive
   hyperparameter optimization (including the no-mixup baseline), used
   cross-validation throughout to obtain more trustworthy performance estimates,
   and also analyzed interpolation properties (which requires GED computations).
5. A key motivation of using graph mixup is to deal with data scarcity, i.e.,
   relatively small training set sizes for complex tasks.

In our statistical analysis, we account for our use of these four datasets when
computing standard errors (esp. under A1).

Even adding a single dataset to the study increases cost substantially and,
given the results so far, may provide only limited insight. We are nevertheless
thinking about and willing to add an additional dataset, although it's not
immediately clear which one would likely provide most additional insight
(suggestions are welcome).

We will also provide the source code of our framework: experiments with new
datasets and new mixup methods can then be performed easily.

### FAQ-4: GED computation is hard. Is the method sufficiently efficient?

Yes. We will provide concrete runtime costs in the revised study (in fact, some
of the prior methods are more costly than GED-Mixup).

While the worst-case time complexity of GED appears impractical, its computation
can be feasible in practice and, in fact, it was feasible in our study. The
development of high-performing methods for exact and approximate GED computation
is an ongoing research direction; e.g., Blumenthal et al. (2020), Chang et al.
(2023), Xu & Chang (2025). These directions are orthogonal to our work.

In our work, we used exact GED computation because we did not want the analysis
to be biased by approximation errors. Using a state-of-the-art exact method, we
computed GEDs for graphs with more than 500 vertices and more than 1000 edges in
our study, without running into computational issues (75% of the GED
computations took less than 0.1s, and these computations are embarrassingly
parallelizable across pairs of graphs). These graph sizes cover a substantial
fraction of the use cases for graph classification.

Finally, note that we use GED computations as well as GED-Mixup as an analysis
tool, not as a method for practical deployment so that runtimes are not as
critical.

---

Blumenthal, D. B., Boria, N., Gamper, J., Bougleux, S., & Brun, L. (2020).
Comparing heuristics for graph edit distance computation. The VLDB Journal,
29(1), 419–458.

Chang, L., Feng, X., Yao, K., Qin, L., & Zhang, W. (2023). Accelerating Graph
Similarity Search via Efficient GED Computation. IEEE Transactions on Knowledge
and Data Engineering, 35(5), 4485–4498. IEEE Transactions on Knowledge and Data
Engineering.

Xu, M., & Chang, L. (2025). Graph Edit Distance Estimation: A New Heuristic and
A Holistic Evaluation of Learning-based Methods. Proc. ACM Manag. Data, 3(3),
167:1-167:24.

---

### Author Response · Authors · 2025-11-25
**Updates on Revision (Part 2)**

## Computational costs (FAQ-4)

We report on the computational cost of mixup below. In short, GED-Mixup was
feasible for its intended purpose of analysis, even with the exact GED
computations that we used.

**Setup:**

When running the experiments on the new datasets NCI1 and IMDB-MULTI, we
maintained runtime statistics (wall clock time) per epoch. We report three
values per method: (i) the per-epoch cost without mixup (baseline), (ii) the
per-epoch cost with a naive application of mixup (naive), and (iii) the
amortized per-epoch cost with an optimized application of mixup (optimized). In
our experiments, we used option (iii) whenever applicable.

In more detail, (ii) naive generates the required mixup graphs individually for
every epoch. As can be seen below, this can add substantial overhead, e.g., for
GeoMix and GED-Mixup. Approach (iii), in contrast, pre-computes a large number
of mixup graphs upfront and samples from these graphs during every epoch; we
report the pre-computation cost amortized over all training epochs. This
optimized approach is not novel and has been used in prior work as well. The
optimized approach is applicable, whenever the mixup is not learned/does not
depend on the training process (i.e., for FGW-Mixup, GeoMix, If-Mixup, SubMix,
GED-Mixup). We will provide more details in the revised version of our paper.

**Results:**

| Method   | Baseline (s) | Naive (s) | Optimized (s) |
| :------- | -----------: | --------: | ------------: |
| Baseline |         0.71 |      0.71 |            NA |
| Emb-M.   |         0.71 |      0.55 |            NA |
| G-Mixup  |         0.71 |     16.43 |            NA |
| GED-M.   |         0.71 |   4813.62 |          1.06 |
| GeoMix   |         0.71 |  16709.86 |          1.73 |
| If-Mixup |         0.71 |      1.57 |          0.99 |
| S-Mixup  |         0.71 |      9.34 |            NA |
| SubMix   |         0.71 |      7.23 |          1.07 |

**Limitations:**

These experiments were conducted on a compute cluster with different kinds of
GPUs and CPUs (see appendix), so that results are somewhat noisy.

## Impact of label noise (FAQ-2)

We conducted experiments on label noise as discussed in FAQ-2. We found that
none of the mixup methods performed better or worse (with statistical
significance) than using no mixup.

**Experimental setup:**

We focused on GIN and IMDB-Binary, and evaluated label corruption probabilities
of \{ 0, 0.125, 0.25, 0.5 \} during training (but not during testing, of
course). To keep costs controlled, we re-used the clean-data hyperparameters
from our study.

**Results:** (classification accuracy (%) ± standard error (%))

| Model | Dataset     | Method   |      0       |    0.125     |     0.25     |     0.5      |
| :---- | :---------- | :------- | :----------: | :----------: | :----------: | :----------: |
| GIN   | IMDB-BINARY | Baseline | 70.77 ± 0.53 | 69.43 ± 1.08 | 66.80 ± 1.51 | 48.27 ± 2.40 |
| GIN   | IMDB-BINARY | Emb-M.   | 67.77 ± 1.89 | 69.07 ± 1.75 | 68.53 ± 0.93 | 48.33 ± 1.64 |
| GIN   | IMDB-BINARY | G-Mixup  | 65.90 ± 2.38 | 67.27 ± 2.05 | 68.07 ± 1.06 | 45.33 ± 2.21 |
| GIN   | IMDB-BINARY | GeoMix   | 70.53 ± 0.60 | 68.17 ± 0.90 | 65.20 ± 1.54 | 47.40 ± 1.32 |
| GIN   | IMDB-BINARY | If-Mixup | 69.30 ± 0.72 | 67.90 ± 1.67 | 67.30 ± 1.54 | 47.27 ± 1.46 |
| GIN   | IMDB-BINARY | S-Mixup  | 69.29 ± 2.16 | 68.33 ± 1.63 | 66.87 ± 0.93 | 48.60 ± 1.53 |
| GIN   | IMDB-BINARY | SubMix   | 70.40 ± 0.45 | 69.07 ± 1.32 | 65.73 ± 1.88 | 47.40 ± 1.01 |
| GIN   | IMDB-BINARY | GED-M.   | 70.40 ± 0.76 | 65.67 ± 2.26 | 63.20 ± 2.32 | 47.23 ± 1.58 |

**Limitations:**

Only a single model, only a single dataset, no separate HPO. Nevertheless, the
results are not promising.

---

### Author Response · Authors · 2025-11-25
**Updates on Revision (Part 1)**

# Updates on revision

We conducted additional experiments along the lines suggested by the reviewers.
We plan to include these results into the revision, but also greatly appreciate
any initial feedback that you may have.

## More datasets (FAQ-3)

We added two further datasets to our empirical evaluation, as discussed in
FAQ-3. We provide results below, including an updated pooling analysis.

In line with our previous results, none mixup method improved over the no-mixup
baseline (see first table below). The conclusions from the pooling analysis
still hold as well (see second table), but two methods now perform significantly
worse than using no mixup: S-Mixup (under A2, was p=0.08, now p=0.01) and
G-Mixup (under A3; was p = 0.69, now p = 0.03).

**Choice of datasets:**

Our previous choice of datasets included a variety of domains already: social
networks (IMDB-BINARY), bioinformatics (ENZYMES, PROTEINS), molecular data
(MUTAG). We have added one further social network dataset (IMDB-MULTI) and one
further molecular dataset (NCI1) so that we now consider two datasets per
domain. We chose these two additional datasets based on their frequency of use,
i.e., both NCI1 and IMDB-MULTI belong to the most frequently used datasets that
we had not already considered.

**Results:** (classification accuracy (%) ± standard error (%))

| Model | Method   | IMDB-MULTI   | NCI1         |
| :---- | :------- | :----------- | :----------- |
| GCN   | Baseline | 46.31 ± 0.60 | 80.35 ± 0.57 |
|       | Emb-M.   | 43.18 ± 1.03 | 81.63 ± 0.29 |
|       | G-Mixup  | 45.73 ± 0.92 | 80.83 ± 0.36 |
|       | GeoMix   | 44.40 ± 0.47 | 80.98 ± 0.31 |
|       | If-Mixup | 45.47 ± 0.73 | 81.43 ± 0.36 |
|       | S-Mixup  | 40.58 ± 1.72 | 79.39 ± 2.44 |
|       | SubMix   | 45.20 ± 1.03 | 81.48 ± 0.38 |
|       | GED-M.   | 46.16 ± 0.76 | 81.58 ± 0.25 |
| GIN   | Baseline | 48.09 ± 0.58 | 81.65 ± 0.42 |
|       | Emb-M.   | 48.22 ± 0.64 | 81.96 ± 0.42 |
|       | G-Mixup  | 47.02 ± 1.35 | 80.84 ± 0.57 |
|       | GeoMix   | 47.49 ± 0.77 | 81.74 ± 0.40 |
|       | If-Mixup | 47.91 ± 0.57 | 81.74 ± 0.34 |
|       | S-Mixup  | 46.84 ± 0.70 | 79.25 ± 1.30 |
|       | SubMix   | 47.96 ± 0.60 | 82.21 ± 0.44 |
|       | GED-M.   | 47.89 ± 0.88 | 82.04 ± 0.48 |

**Results:** (Accuracy (%), pooled standard errors (pp), $p$ values under
assumptions (A1)-(A3); $p < 0.05$ in **bold**)

| Method   | Accuracy | SE (A1) | p (A1) | SE (A2) | p (A2)   | SE (A3) | p (A3)   |
| :------- | :------- | :------ | :----- | :------ | :------- | :------ | :------- |
| Baseline | 70.29    | 1.04    | NA     | 0.46    | NA       | 0.30    | NA       |
| Emb-M.   | 69.49    | 1.07    | 0.60   | 0.43    | 0.21     | 0.30    | 0.06     |
| G-Mixup  | 69.41    | 1.07    | 0.56   | 0.48    | 0.19     | 0.29    | **0.03** |
| GeoMix   | 69.11    | 1.10    | 0.44   | 0.54    | 0.10     | 0.27    | **0.00** |
| If-Mixup | 70.58    | 1.06    | 0.84   | 0.40    | 0.63     | 0.25    | 0.44     |
| S-Mixup  | 68.11    | 1.30    | 0.19   | 0.66    | **0.01** | 0.49    | **0.00** |
| SubMix   | 70.69    | 1.07    | 0.79   | 0.42    | 0.52     | 0.24    | 0.29     |
| GED-M.   | 70.54    | 1.04    | 0.86   | 0.42    | 0.68     | 0.19    | 0.46     |

---

### Author Response · Authors · 2025-12-03
**Revision summary (part 2)**

(continues from the first part)

## GED computation is hard. Is the method sufficiently efficient?

_Preliminary details_: FAQ-4

_Raised by:_ Reviewers uWeG W4, z4K1 W3/Q2

_Our steps:_

- We added further information on run time cost for GED-Mixup, but also for all
  other mixup methods considered in this study (C.3). The results suggest that
  GED-Mixup was sufficiently efficient to be feasible, but also that graph mixup
  in general may lead to substantial computational overhead.

## Other points raised in the reviews

> bfwX W2. The discussion of the statistical assumptions A1-A3 for pooled
> analysis were difficult to follow, and it was not always clear when which
> assumption applied

We now expanded the discussion (Sec. 3, pooled analysis) to provide a more
high-level overview and generally improved presentation of this section. We now
also clearly indicate the assumptions used in each of the tables/figures, added
more results (Fig. 2), and provided all p-values (Tab 9, Tab 10).

> uWeG W5 (stated as "as a suggestion"). There are newer methods (MomentMixup,
> SIGL).

We now briefly mention these methods in Sec. 2 and Tab. 1, and clearly state
that they are not included in our experimental study and why (footnote 1). Both
methods were published concurrently to our work, they output graphons (as
G-Mixup, which we did consider), and they may require substantial cost to
evaluate. That being said, we consider the evaluation of newer methods valuable
and we will provide source code for all of our experiments so that such an
evaluation can be performed.

> CTcZ Q2. Is the link between interpolation fidelity and downstream accuracy
> dependent on domain semantics, such as chemically meaningful edit operations?

We now added a discussion around this point (A.3), to which we agree. Domain
knowledge would clearly be beneficial, as would be automated methods which
"learn" meaningful edit operations.

---

### Author Response · Authors · 2025-12-03
**Revision summary (part 1)**

Dear reviewers, dear area chair,

We were in the midst of discussions with reviewers when the reviewing procedure
needed to be adjusted. We've since then revised the paper, aiming to address
each of the suggestions raised in the reviews as thoroughly as possible without
further feedback.

We feel that---whether or not our work ultimately will be accepted to ICLR---the
paper has improved substantially. We thank all of you for your valuable support
of our work.

The original ratings of the reviewers were (in increasing order): 4, 4, 6, 8.
None of the reviewers had updated their reviews or ratings before reviewer
feedback was frozen; we were still discussing their feedback and thoughts.

In what follows, we summarize the thoughts and suggestions made by reviewers,
and briefly describe how we addressed them in the revision. All changes are
marked in blue in the revised paper.

## Mixup methods with bad interpolation properties fell behind. What can we learn from good interpolation properties though?

_Preliminary details_: FAQ-1

_Raised by:_ Reviewers BfwX W1/Q1/Q2, uWeG W1, CTcZ W2/W3, z4K1 W1/W2

- Generally, we neither believe nor claim that graph mixup cannot be beneficial.
  The key takeway of our study is that there is currently insufficient empirical
  evidence, both for and against graph mixup, and we also explore why this is
  the case. We clarified these points in the revision; see details below.
- We carefully revised the paper to very clearly state that bad interpolation
  properties were detrimental in our study, but good interpolation properties on
  their own are not sufficient (R4 in paper).
- We added a new appendix (A.3) exploring this question, arguing why good
  interpolation may not be enough and giving directions for future research.
- We added a new appendix (A.2), arguing that---at least in principle---bad
  interpolation properties may still lead to a useful training signal. The goal
  of graph mixup, however, is to interpolate between inputs so that we do not
  consider such methods as mixup methods.
- Reviewer CTcZ also suggested to explore the impact of interpolation error in a
  controlled environment. This was a valuable suggestion and we conducted such
  an experiment (Tab. 5, C.4). In our point of view, the results led to further
  evidence for the main takeaways of our study.

## The paper focuses on classification accuracy. What about other tasks or settings?

_Preliminary details_: FAQ-2

_Raised by:_ Reviewers bfwX W1, uWeG W2, cTcZ W2, z4K1 W1

More concretely, the reviewers suggested that graph mixup may also be beneficial
w.r.t. graph-level regression, robustness, distribution shifts, topology
perturbations, label noise, data imbalance, and that controlled experiments may
help.

- We added such a controlled experiment (discussed above; Tab. 5, C.4),
  providing further evidence for the main takeaways of our study.
- We conducted an additional experiment to study robustness under label noise
  (App C.5). We did not find supporting evidence for graph mixup in this
  experiment.
- We used more datasets (Tab. 6) and expanded the dataset descriptions (C.1).
  Some of the datasets indeed already exhibit data imbalance.
- We did not explore whether graph mixup may or may not be beneficial in case of
  distribution shifts or topology shifts, and neither did prior work on graph
  mixup. We do consider this point valuable though. The scope of our experiments
  is already substantial, and it's not always feasible to answer all possible
  questions. As stated above, we carefully revised the paper so that we do not
  claim that graph mixup may not be beneficial. We also added additional
  thoughts on this point (A.1, A.3).
- The focus of our study is on graph classification, as this is the setting most
  often explored in prior work. We clearly state this focus, starting right in
  the abstract. We do, however, discuss selected regression problems in A.1,
  A.2, and A.3 (new); each regression problem can be turned into a graph
  classification problem (e.g., by quantizing output ranges into buckets such as
  "less than 0" and "larger than 0").

## The study uses four real-world datasets. What about other datasets?

_Preliminary details_: FAQ-3

_Raised by:_ Reviewer uWeG W3, CTcZ W1/Q1, z4K1 W1

- We added two additional datasets to our experiments (see Tab. 2 and Tab. 4).
  Our full set of datasets now covers relevant domains (bioinformatics, social
  networks, molecules), each with two datasets. The additional experiments
  further supported our previous findings.
- We expanded the dataset descriptions (C.1) to more clearly contrast properties
  of these datasets.
- We included the rationale for choice of datasets (C.1), clearly arguing why we
  included the datasets used in our study, and why we did not consider some
  other datasets (even though, of course, using more datasets always provides
  further evidence).

(continues in second post)

---

### Meta-Review · Area_Chair_eK2F · 2026-01-06

**Summary:**

While the paper presents a thorough and methodologically sound evaluation of graph mixup methods, we find that its overall contribution is incremental and its practical impact is limited. The study convincingly demonstrates that current graph mixup approaches fail to yield statistically significant improvements, and it offers a useful diagnostic perspective via interpolation error analysis. However, the work remains largely critical in nature—it identifies shortcomings without proposing a novel, viable alternative or a clear path forward. The proposed GED-Mixup, while insightful as an analytical tool, is computationally prohibitive for real-world use and does not represent a practical advance. Given the emphasis in our venue on significant conceptual, algorithmic, or empirical innovations, we feel that the paper does not meet the bar for acceptance in its current form.

**Reviewer Scores:**

No

---

### Decision · Program_Chairs · 2026-01-26

Reject